# scPrediXcan integrates deep learning methods and single-cell data into a cell-type-specific transcriptome-wide association study framework

## Graphical abstract

## Authors

Yichao Zhou, Temidayo Adeluwa, Lisha Zhu, ..., Boxiang Liu, Mengjie Chen, Hae Kyung Im

## Correspondence

mengjiechen@uchicago.edu (M.C.), haky@uchicago.edu (H.K.I.)

## In brief

Zhou et al. introduce scPrediXcan, a novel transcriptome-wide association study framework that integrates the deep learning-based model ctPred for cell-type-specific expression prediction. Applied to type 2 diabetes and lupus, scPrediXcan outperforms canonical TWAS methods with higher power and more discoveries, providing new insights into disease mechanisms at the cellular level.

## Highlights

- A deep learning model, ctPred, for cell-type-specific gene expression prediction

- A new TWAS framework, scPrediXcan, to perform TWAS at the cell-type level

- ScPrediXcan outperforms canonical TWAS frameworks in T2D and SLE with higher power

 Zhou et al., 2025, Cell Genomics 5, 100875
May 14, 2025 © 2025 The Authors. Published by Elsevier Inc.

CellPress

## Article

# scPrediXcan integrates deep learning methods and single-cell data into a cell-type-specific transcriptome-wide association study framework

Yichao Zhou,[1] Temidayo Adeluwa,[1] Lisha Zhu,[2] Sofia Salazar-Magaña,[2] Sarah Sumner,[2] Hyunki Kim,[3] Saideep Gona,[1] Festus Nyasimi,[4] Rohit Kulkarni,[3] Joseph E. Powell,[5,6] Ravi Madduri,[7] Boxiang Liu,[8] Mengjie Chen,[2,*] and Hae Kyung Im[2,9,*]

[1]Committee of Genetic, Genomics, and Systems Biology, University of Chicago, Chicago, IL 60637, USA
[2]Section of Genetic Medicine, Department of Medicine, University of Chicago, Chicago, IL 60637, USA
[3]Department of Medicine, Harvard Medical School, Boston, MA 02115, USA
[4]Department of Human Genetics, University of Chicago, Chicago, IL 60637, USA
[5]UNSW Cellular Genomics Futures Institute, University of New South Wales, Sydney, NSW 2052, Australia
[6]Translational Genomics, Garvan Institute of Medical Research, Sydney, NSW 2010, Australia
[7]Data Science and Learning Division, Argonne National Laboratory, Chicago, IL 60439, USA
[8]Department of Pharmacy and Pharmaceutical Sciences, National University of Singapore, Singapore 119077, Singapore
[9]Lead contact
*Correspondence: mengjiechen@uchicago.edu (M.C.), haky@uchicago.edu (H.K.I.)

## SUMMARY

Transcriptome-wide association studies (TWASs) help identify disease-causing genes but often fail to pinpoint disease mechanisms at the cellular level because of the limited sample sizes and sparsity of cell-type-specific expression data. Here, we propose scPrediXcan, which integrates state-of-the-art deep learning approaches that predict epigenetic features from DNA sequences with the canonical TWAS framework. Our prediction approach, ctPred, predicts cell-type-specific expression with high accuracy and captures complex gene-regulatory grammar that linear models overlook. Applied to type 2 diabetes (T2D) and systemic lupus erythematosus (SLE), scPrediXcan outperformed the canonical TWAS framework by identifying more candidate causal genes, explaining more genome-wide association study (GWAS) loci and providing insights into the cellular specificity of TWAS hits. Overall, our results demonstrate that scPrediXcan represents a significant advance, promising to deepen our understanding of the cellular mechanisms underlying complex diseases.

## INTRODUCTION

Transcriptome-wide association studies (TWASs) are a class of methods that nominate candidate causal genes for complex traits and diseases by determining associations between predicted gene expression and phenotype.[1–3] Canonical TWAS approaches train a gene expression prediction model using tissue-level gene expression from a reference panel of at least 100 individuals. While TWASs have been successfully applied to various tissues and traits, providing candidate causal gene lists,[4,5] they are limited by the mismatch between available expression panels and disease-relevant cell types or states. Tissues with extensive expression panels (e.g., whole blood or lymphoblastoid cell lines) are commonly employed to maximize power, but recent studies suggest that context-specific regulation of gene expression is more relevant for disease.[3,6,7] This is especially true when rare cell types that are underrepresented in conventional bulk tissue expression data drive disease onset. Therefore, a TWAS framework with a model that can predict expression from disease-relevant tissues and/or cell types

would be of great value for uncovering genes involved in the trait etiology. Although our understanding of cellular heterogeneity and cell-type-specific regulatory patterns has improved dramatically with recent advances in single-cell RNA sequencing (scRNA-seq),[8] single-cell data are simply not available on the scale required by canonical TWAS frameworks to ensure optimal accuracy and power across contexts.

Recent advancements in deep learning models that predict molecular features based on sequence data present a promising solution. Models predicting molecular features (e.g., epigenome) are trained using only a reference genome, eliminating the need for population-level data.[9–11] However, existing deep learning models for gene expression predictions are trained on bulk RNA-seq data and are constrained to tissue-level predictions. Moreover, directly training a sequence-to-expression model for cell-type-specific prediction remains challenging due the high sparsity of the scRNA-seq data. To overcome this, we developed a cell-type-specific gene expression prediction model, ctPred, which leverages pseudo-bulk scRNA-seq data and employs a pre-trained sequence-to-epigenomics model, Enformer.

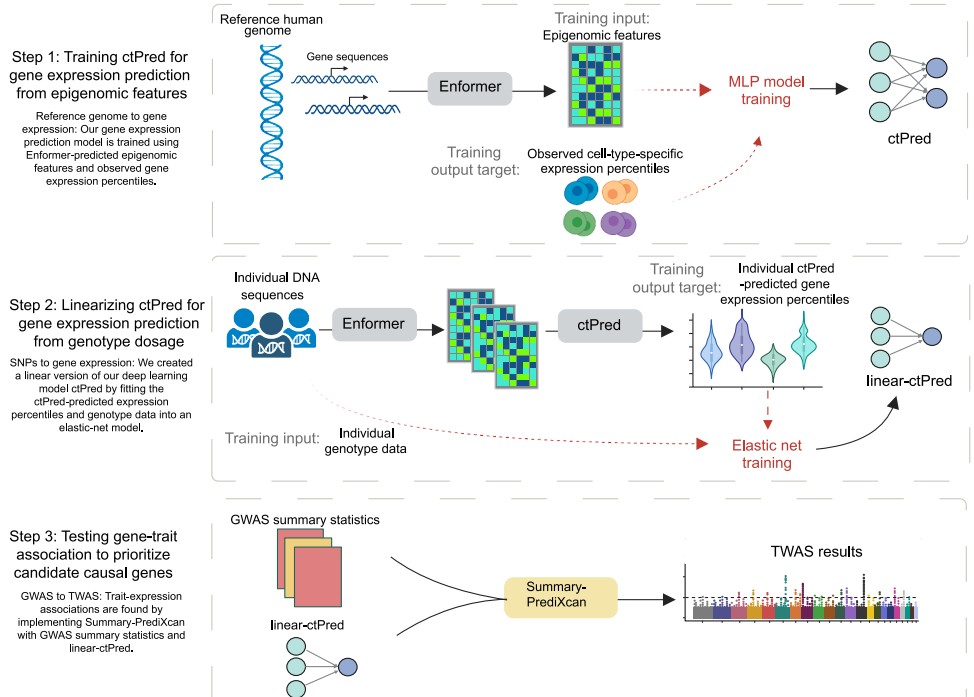

**Figure 1. Overview of the scPrediXcan framework**

By using Enformer as a feature extractor, ctPred enhances convergence and improves overall performance. We further developed single-cell PrediXcan (scPrediXcan), a cell-type-level TWAS framework, by leveraging ctPred to perform TWASs using single-cell data. While ctPred is theoretically capable of predicting context-specific, *in silico* expression data at the individual level for TWASs using genome-wide association study (GWAS) data, the computational demands remain prohibitive given the current scale of GWASs (i.e., hundreds of thousands of individuals). Additionally, access to individual-level GWAS data is often difficult to obtain. To overcome these challenges, we introduced a SNP-based linear version of ctPred, termed $\ell$-ctPred, which is derived from genotype data alongside a ctPred-predicted, *in silico* expression reference panel. Using $\ell$-ctPred's weights, we can conduct association tests between genes and diseases using only summary statistics from GWAS data, enabling a TWAS that is fundamentally based on single-cell data.

We evaluated scPrediXcan by applying it to two diseases, type 2 diabetes (T2D) and systemic lupus erythematosus (SLE). Our comparison of scPrediXcan's performance against canonical TWAS models trained using the same datasets reveals that scPrediXcan significantly outperforms the canonical frameworks. It identifies a larger number of candidate causal genes, explains more GWAS loci, and provides more detailed insights into the cellular specificity of the TWAS findings. These results highlight scPrediXcan's substantial improvement in both the accuracy and relevance of gene nominations within TWASs, demonstrating its promising potential to advance our understanding of the cellular mechanisms that underpin complex diseases.

## RESULTS

### Overview of the scPrediXcan framework

The scPrediXcan framework consists of three key steps (Figure 1; Table S47). First, we trained a model to predict gene expression percentiles from epigenomics data and observed cell-type-specific gene expression. Second, we linearized this deep learning model into a SNP-based elastic net model, which can be used for association tests using GWAS summary statistics. Finally, we tested associations between genes and the trait of interest.

In the first step, we established a method, ctPred, to predict the cell-type-specific gene expression from individual DNA sequences. Because it is challenging for the model to learn genomic grammar by training directly on the highly sparse single-cell expression data, we utilized the genomic regulation insights gained from the state-of-the-art sequence-to-epigenomics model trained on bulk-level data, Enformer.[9] Enformer was trained and tested on human and mouse genomic sequences. When used for prediction in humans, Enformer takes a 196,608-bp sequence of DNA as input (which we centered at each gene's transcription start site) and predicts a $5,313 \times 896$ output matrix. The 5,313 epigenomic features—comprising transcription factors, chromatin immunoprecipitation sequencing, histone modifications, DNase sequencing or assay for transposase-accessible chromatin using sequencing, and cap analysis of gene expression profiles—are aggregated into 128-bp bins. We used Enformer as a feature extractor; we used the genetically determined epigenomic features from Enformer to train ctPred—a lightweight, four-layer multi-layer perceptron (MLP) that generates single-cell expression data as output.

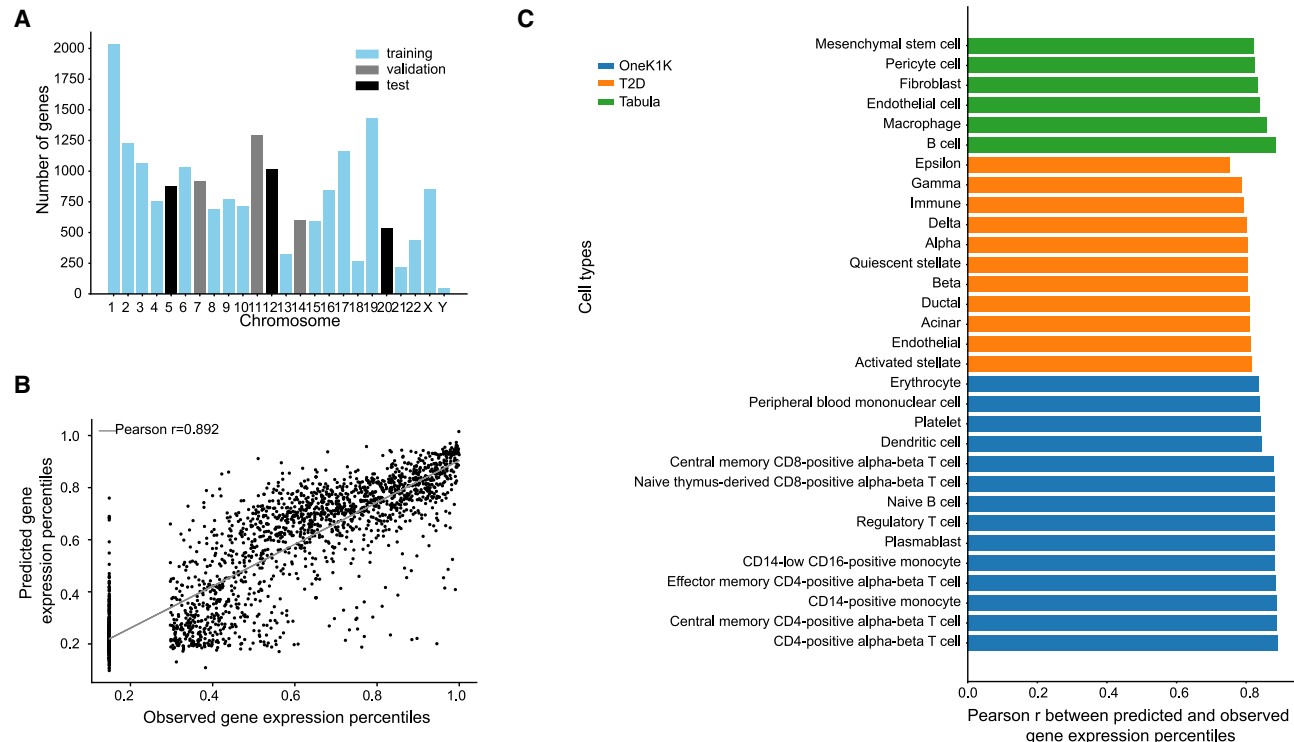

**Figure 2. ctPred predicts cell-type-specific gene expressions across the genome**
(A) Data splitting by chromosomes for ctPred training.
(B) Scatterplot of the predicted gene expression percentiles and observed gene expression percentiles for each gene in a CD4$^+$ α-β T cell from the OneK1K dataset.
(C) Bar plot of Pearson correlations between predicted and observed gene expression percentiles in various cell types from the OneK1K, T2D, and Tabula Sapiens subsets. Results for all cell types are listed in Table S3.

Despite ctPred's ability to predict individual cell-type-specific gene expression from gene sequences, the computational expense of such predictions for TWAS association tests across large GWAS cohorts remains high, and access to individual-level data is limited. We addressed this in our workflow's second step by transforming the deep learning model into a linear form, creating a SNP-based elastic net version of ctPred, termed ℓ-ctPred. This version can predict gene expression for specific cell types from SNP dosages. The weights for ℓ-ctPred are stored in a database, eliminating the need for end users to repeat the training steps.

The final step of our workflow tests for associations between genes and traits or diseases at the cell type level using S-PrediXcan.[12] This step employs the weights from ℓ-ctPred along with GWAS summary statistics to estimate gene effect sizes on the trait and to compute $p$ values, prioritizing putative causal genes.

### ctPred accurately predicts single-cell pseudobulk gene expression across the genome in diverse cell types and datasets

To develop prediction models for different cell contexts and evaluate their performance, we trained ctPred and tested its prediction performance on three scRNA-seq datasets separately: the OneK1K dataset,[13] a T2D islet dataset, and the Tabula Sapiens

dataset.[14] The OneK1K dataset includes 29 cell types from 982 individuals, the T2D islet dataset has 11 cell types from 29 individuals, and the Tabula Sapiens dataset contains more than 150 cell types from 14 different organs of 15 individuals.

For each dataset, we tailored ctPred models to individual cell types. To avoid data leakage due to sequence overlaps between genes, we randomly divided the datasets by chromosomes into training, validation, and test sets (Figure 2A). We then evaluated the models' performance by calculating Pearson correlations between predicted and observed pseudobulk gene expression percentiles (STAR Methods) across the test sets (Figure 2B). For example, in the OneK1K dataset, Pearson correlations for all cell types exceed 0.8, with the highest at 0.892 in CD4$^+$ α-β T cells and the lowest at 0.836 in erythrocytes (Figure 2C). indicating that ctPred accurately predicts gene expression across diverse cell types.

Further testing the model's robustness, we validated ctPred on 11 cell types from the T2D dataset and 178 cell types from the Tabula Sapiens subset (Table S46); we show results from the T2D dataset and six representative cell types from Tabula Sapiens in Figure 2C. In the T2D dataset, Pearson correlations range from 0.753 in epsilon cells to 0.815 in activated stellate cells. In the selected Tabula Sapiens cell types, correlations range from 0.823 in mesenchymal stem cells to 0.885 in B cells, showcasing the model's effectiveness across various datasets and cellular

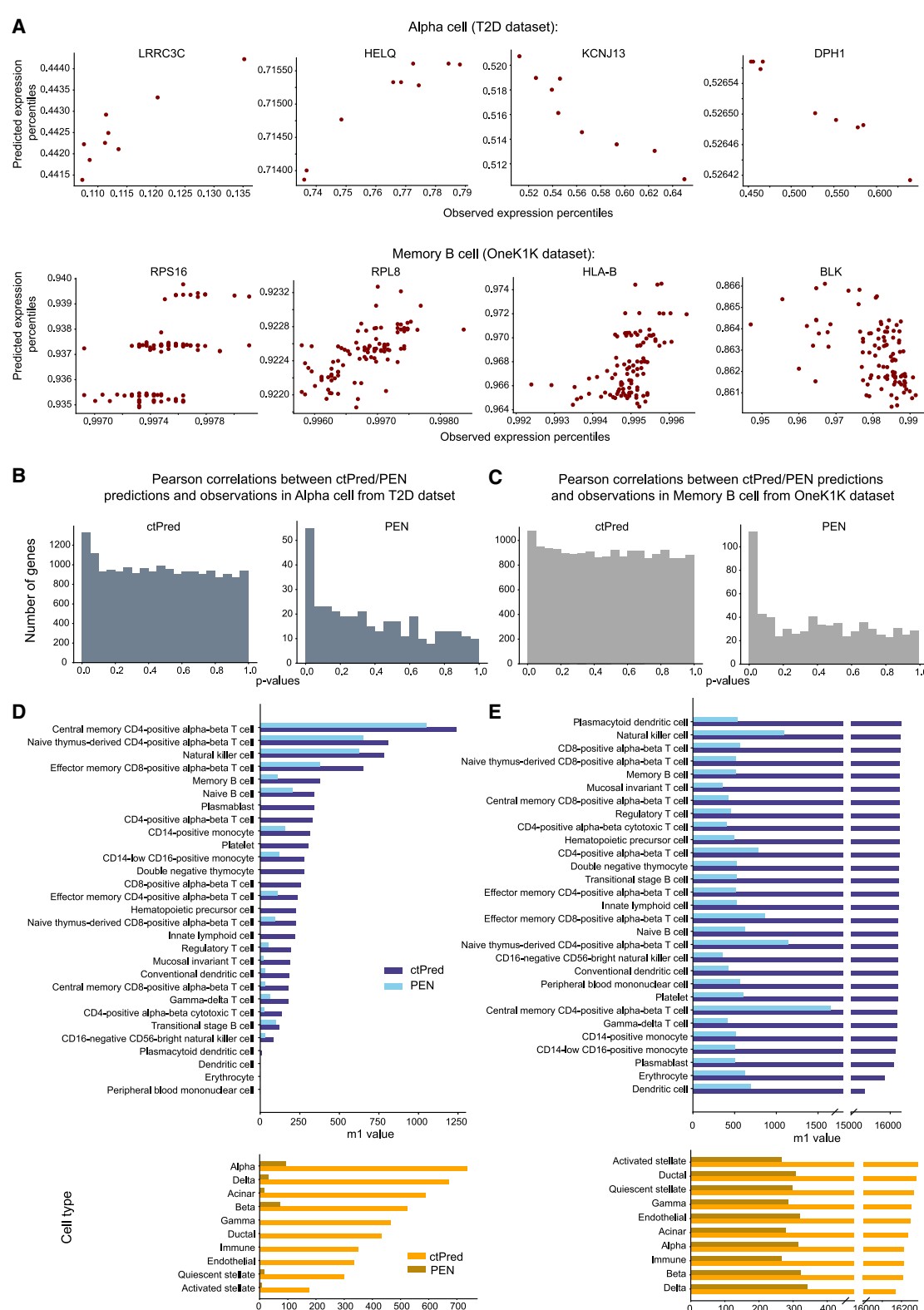

**Figure 3. ctPred predicts cell-type-specific gene expressions across individuals**

(A) Scatterplot of ctPred-predicted and observed gene expression percentiles for representative genes missed by PEN in α cells and memory B cells in the test set.

*(legend continued on next page)*

contexts. To investigate the impact of epigenomic features on ctPred's model predictions, we conducted Shapley value analysis on ctPred trained on T cells from the Tabula Sapiens dataset (Figure S9). Among the top 10 most impactful features, seven were immune related. Furthermore, across all epigenomic features, T cell-specific features exhibited higher Shapley additive explanation or SHAP values (i.e., absolute impact) on model predictions compared to non-T cell features. This finding highlights that ctPred prioritizes cell-type-specific epigenomic features to enhance the accuracy of cell-type-specific gene expression predictions.

During our project, we learned of the parallel development of a module within seq2cells[15] called emb2cell, which specifically focuses on single-cell pseudobulk gene expression prediction. Similar to our ctPred, emb2cell uses embeddings for prediction, but while ctPred utilizes the 5,313-dimension output from Enformer, emb2cell leverages 3,072-dimension intermediate embeddings to train a two-layer MLP model. A key distinction between the models is that ctPred is much more parameter efficient, containing approximately 0.4 million parameters compared to emb2cell's over 60 million. We evaluated both models using CD4[+] T cell scRNA-seq data, adhering to the training, validation, and testing partitions specified in the seq2cells publication. ctPred achieves a Pearson correlation of 0.787 across genes (Figure S1), significantly outperforming the 0.666 correlation reported for emb2cell.

The minimum number of cells per cell type was 125 in OneK1K, 212 in T2D, 4,539 in the Tabula Sapiens subset we selected as representative cell types, and 2 in the whole Tabula Sapiens dataset. In general, prediction performance improved with an increasing number of available cells within each cell type, underscoring the benefits of our approach, where reads are aggregated across individuals (Figures S6A, S6B, S6E, S6F, and S6I). This advantage becomes even more evident when we assess performance across individuals.

Overall, across 46 cell types and three datasets, ctPred not only achieves Pearson correlations ranging from 0.753 to 0.892 but also surpasses emb2cell on the CD4[+] T cell dataset despite having 150 times fewer parameters. This underscores ctPred's efficiency, accuracy, and robustness in predicting cell-type-specific gene expression.

### ctPred outperforms SNP-based predictors used in canonical TWASs for predicting cell-type-level gene expression across individuals

To assess ctPred's performance across individuals—crucial for TWAS aiming to discern gene expression changes between patients and healthy controls—we compared it with a SNP-based approach used in canonical TWASs. This SNP-based method, which we refer to as pseudobulk elastic net (PEN), is trained on

the same observed single-cell data as ctPred and should not to be confused with ℓ-ctPred, which also uses elastic net but is based on *in silico* gene expression predictions.

We found that PEN fails to converge for the majority of genes attempted, yielding predictors for only ∼3.5% of expressed genes, whereas the ctPred approach yields predictors for all expressed genes (∼19,000). Specifically, PEN yields 318–434 (median = 354) and 504–1784 (median = 707) across cell types in T2D and OneK1K, respectively. Figure 3A provides examples of ctPred-predicted versus observed gene expression percentiles, highlighting some genes missed by the PEN model.

Next, we assessed the correlation between the predicted and observed gene expression across individuals to evaluate predictor performance. Due to the known correlation sign inconsistency in Enformer,[16,17] we utilized the $p$ value of correlation as the primary performance metric (Discussion). The histograms of $p$ values for correlations in α cells and memory B cells are shown in Figures 3B and 3C. However, given the substantial difference in the number of genes predicted by each model, caution must be taken to ensure a fair comparison. Since convergence and prediction performance are intertwined, focusing solely on genes that converge in PEN would introduce bias. One robust metric is the number of "true positive genes" (i.e., m1 = $\pi_1$* #genes, where $\pi_1$ is the proportion of true positive genes), estimated using the q value framework,[18] and represents genes with $p$ values deviating from a uniform distribution (Figure S3). Our ctPred approach consistently identifies a larger number of true positive genes compared to the traditional PEN method when validated against observed expression, though the total number of genes conclusively associated with observed expression is modest, ranging from 0 to 1,236 with a median of 275 (Figure 3D).

Since our primary objective is to elucidate how GWAS variants influence phenotypes through the regulation of molecular phenotypes like cell-type-specific gene expression (i.e., genetically regulated expression [GreX]), we also evaluated the performance of ctPred and PEN using predictors trained on geneotype-tissue expression (GTEx) bulk data[19] as proxies for the genetically regulated component of expression (GTEx GReX). We predicted gene expression in 400 European individuals using 1000G cohort[20] genotype data and GTEx-trained weights, which we compared against both ctPred and PEN predictions in the same cohort. We calculated correlations, derived $p$ values, and estimated $\pi_1$ to assess the proportion of genes for each model genuinely associated with GTEx GReX proxies. This analysis primarily identifies associations between components of GReX shared across cell types and bulk tissues, suggesting that the ability to predict shared regulation may indicate potential in predicting cell-type-specific components despite data limitations for direct testing. We evaluated the correlation of our cell

(B) Histogram of the distribution of Pearson correlation $p$ values between model-predicted and observed gene expression percentiles in α cells from the T2D dataset.

(C) Histogram of the distribution of Pearson correlation $p$ values between model-predicted and observed gene expression percentiles in memory B cells from the OneK1K dataset.

(D) Bar plot of m1 values (i.e., the number of true-positive genes) for ctPred and PEN in different cell types when compared against observed expression percentiles.

(E) Bar plot of m1 values (i.e., number of true positive genes) for ctPred and PEN in different cell types when compared against GTEx GReX.

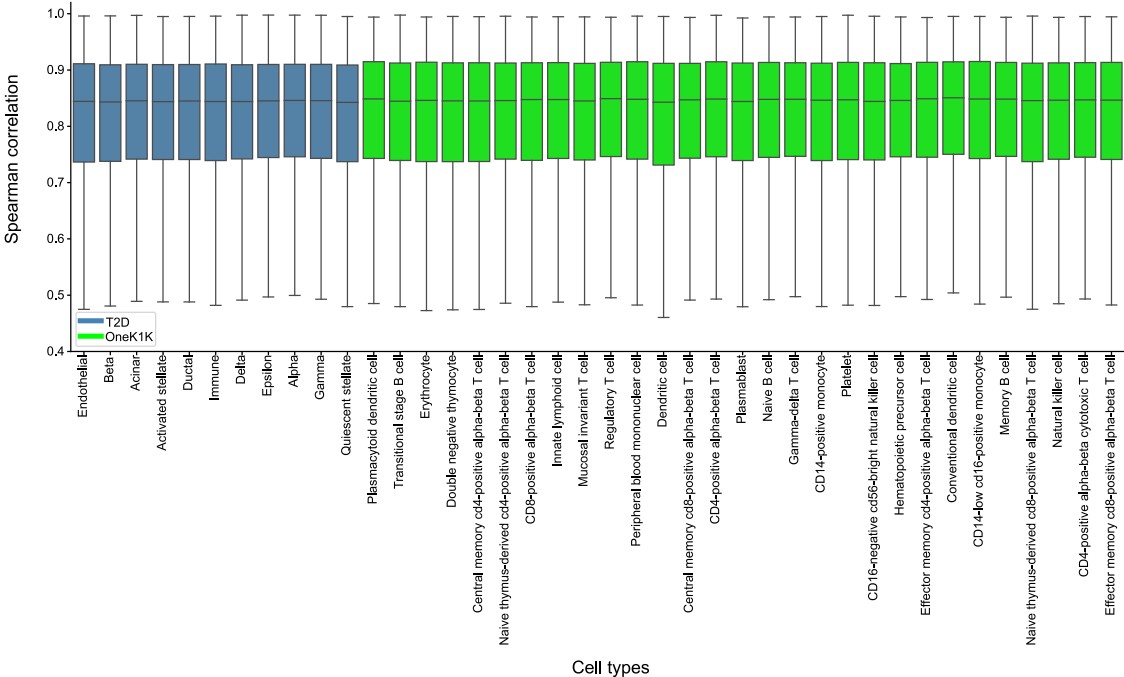

**Figure 4. ℓ-ctPred approximates ctPred with high concordance**
Box plot of Spearman correlations between ctPred predictions and ℓ-ctPred predictions across individuals in 40 different cell types from the T2D and OneK1K datasets. Each box respresents the interquartile range of the data, with the median indicated by the center bar. The whiskers indicate the range of the data for each cell type.

type expression predictors with GTEx GReX across 49 tissues, aggregating results into a single $p$ value per gene per cell type using the aggregated Cauchy association test (ACAT) method.[21] Approximately 19,000 genes were tested using ctPred predictors, while PEN predictors were used to test between 318 and 1,784 genes, constrained by the limited predictors this approach yields. Using the $\pi_1$ statistic, we estimated that 95.3%–96.1% (median = 95.7%) of ctPred-predicted genes and 59.2%– 95.5% (median = 77.8%) of PEN-predicted genes are truly correlated with GTEx GReX. The disparity between model performance is even more stark when considering the number of truly correlated genes: ctPred predicts 15,339–16,277 (median = 16,206) truly correlated genes, while PEN only predicts 277– 1,646 (median = 458) (Figure 3E).

These findings confirm that ctPred more effectively predicts shared regulation than PEN and suggests that ctPred may also better predict cell-type-specific regulation. Given ctPred's robust performance, we decided to include all genes predicted by ctPred in our phenotype association tests. According to a recent analysis,[22] incorporating genes not associated with GReX does not compromise the type I error rate, provided that the appropriate adjustments for multiple testing are made.

Furthermore, our comparison of performance across individuals and the number of cells per cell type reveals a trend similar to that observed for performance across genes. As shown in Figure S7, the number of true positive genes per cell type increases with the total number of available cells. The canonical TWAS, which relies on variation across individuals, is more sen-

sitive to a low number of cells compared to ctPred, which gains robustness by aggregating cells across individuals.

### Linear ctPred enables large-scale context-specific TWAS

Having developed reliable context-specific prediction models, we are theoretically equipped to conduct TWASs using single-cell informed expression levels. However, the computational burden when using Enformer and ctPred at the scale required is significant; we estimate that predicting gene expression for 500 individuals across 20,000 genes would require ~2,700 graphics processing unit (GPU)-hours. Additionally, individualized data (e.g., raw GWAS data) are difficult to access for most diseases. An efficient alternative is to use readily accessible GWAS summary statistics for association studies. S-PrediXcan is the commonly used method for performing TWASs using GWAS summary statistics[12]; however, it requires linear gene expression predictors, meaning that ctPred is incompatible with the framework. To address this incompatibility, we created an *in silico* reference dataset using ctPred predictions for several hundred individuals and fitted a SNP-based elastic net model to these data. This resulting model, ℓ-ctPred, takes individual genotype data and yields predicted expression levels using linear combinations of SNP dosages (Figure 1, step 2).

We linearized ctPred models for 40 cell types from T2D and OneK1K datasets using genotype data of 462 European individuals from the 1000G project and stored the weights for downstream association analysis. We used European ancestry data

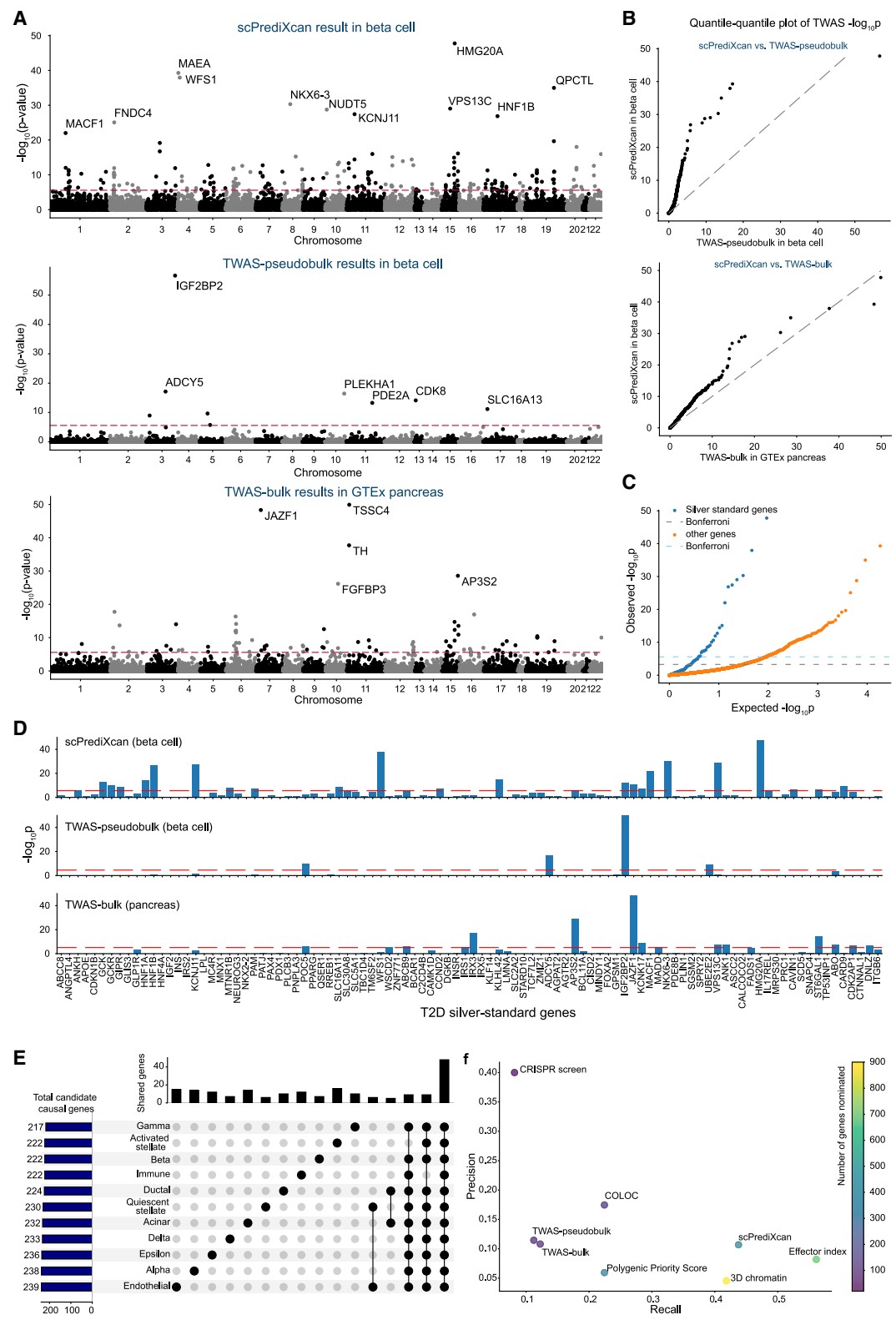

to better match the ancestry of available GWASs. We validated linearization efficiency by calculating the 10-fold cross-validated Spearman correlations between predictions from ctPred and $\ell$-ctPred across all genes (Figure S4). For all cell types, the median correlation is above 0.83, indicating that $\ell$-ctPred robustly approximates ctPred (Figure 4). We also compared the performance of $\ell$-ctPred to PEN in T2D and OneK1K datasets; like ctPred, $\ell$-ctPred outperformed PEN across cell types (Figure S8). Thus, we integrated $\ell$-ctPred as the gene expression predictor in the scPrediXcan framework.

While both the canonical TWAS gene expression predictors and $\ell$-ctPred use a SNP-based elastic net model, $\ell$-ctPred has a significantly higher number of genes converged during the model training. For example, for the memory B cell type from the OneK1K dataset, the canonical SNP-based model trained on observed expression data achieved convergence for 340 genes with a training sample size of 800 individuals and required both genotype and scRNA-seq data. In contrast, $\ell$-ctPred achieved convergence for 16,892 genes with a training sample size of only 462 individuals using solely genotype data. The stark difference in the number of converging genes arises because the canonical TWAS prediction model is based on observed data, whereas $\ell$-ctPred relies on *in silico* ctPred-predicted gene expression data. The *in silico* data contain only the genetic component of expression, while the observed gene expression includes both genetic and environmental components, making $\ell$-ctPred less vulnerable to the noise and sparsity present in observed scRNA-seq data. Furthermore, $\ell$-ctPred is not limited by the small sample sizes associated with observed scRNA-seq data, as it can predict more individual expression as needed from the given genotype data.

## scPrediXcan enables TWAS at the single-cell pseudobulk level for T2D and outperforms canonical TWAS methods

Our context-specific TWAS framework, scPrediXcan, utilizes cell-type-specific $\ell$-ctPred models as the predictive component within the S-PrediXcan framework to explore gene-disease associations at the cell type level (Figure 1, step 3). To demonstrate the efficacy of scPrediXcan in identifying candidate causal genes, we conducted comparisons involving several benchmarks: (1) a cell-type-specific canonical TWAS method (TWAS-pseudobulk) that uses the same scRNA-seq pseudobulk data as scPrediXcan (i.e., PEN predictors), (2) a tissue-level canonical TWAS method (TWAS-bulk) that relies on bulk RNA-seq data, and (3) other gene prioritization methods for T2D (STAR Methods).

scPrediXcan identifies a larger number of candidate T2D-associated genes across more GWAS loci compared to both TWAS-pseudobulk and TWAS-bulk. Specifically, using scPrediXcan, we identified 222 candidate causal genes across 108 different linkage disequilibrium (LD) blocks from a total of 1,703 pre-defined approximately independent LD blocks.[23] In contrast, we identified only 12 candidate genes across 11 LD blocks with TWAS-pseudobulk and 111 genes across 64 LD blocks with TWAS-bulk. Representative results for all three frameworks are shown for β cells (scPrediXcan and TWAS-pseudobulk) and pancreas (TWAS-bulk) in Figure 5A. The full set of association statistics is shown in Tables S4–S14.

Further, we evaluated the statistical significance of these findings by comparing the TWAS *p* of scPrediXcan against those from TWAS-pseudobulk and TWAS-bulk through a quantile-quantile plot (QQ plot; Figure 5B). Considering that $\ell$-ctPred achieves convergence for significantly more genes than the SNP-based models used in the other two frameworks, we used a uniform distribution of *p* values to represent genes absent in the canonical approaches, ensuring a comprehensive comparison. The QQ plot clearly demonstrates that scPrediXcan outperforms the canonical TWAS frameworks, consistently showing statistically lower *p* values for identified TWAS hits.

To further evaluate scPrediXcan's performance compared to other canonical TWAS methods based on bulk data, we compared its results in β cells to those of UTMOST,[24] another canonical TWAS method, applied to pancreatic tissue (Figure S10). scPrediXcan identified slightly more candidate genes than UTMOST (222 vs. 221) but achieved higher precision (0.109 vs. 0.082) and a significantly higher recall/power (0.439 vs. 0.153) in identifying T2D "silver-standard" genes (described below). These results verify that scPrediXcan outperforms canonical TWAS methods in detecting functionally disease-relevant genes.

To evaluate scPrediXcan's efficacy in identifying causal genes for T2D, we utilized a curated list of T2D-associated genes from the Common Metabolic Diseases Knowledge Portal database,[25] which we call T2D silver-standard genes (Table S45). We compared the scPrediXcan *p* values for these genes against those of the remaining genes predicted by $\ell$-ctPred. A QQ plot against a uniform *p* value distribution (Figure 5C) demonstrates that the silver-standard genes have significantly lower *p* values, affirming that scPrediXcan accurately identifies genes truly associated with T2D. We also compared these results with those from canonical TWAS frameworks; we show representative results for β cells (Figure 5D). Among 98 silver-standard genes,

---

**Figure 5. scPrediXcan in T2D outperforms canonical TWAS methods**

(A) Manhattan plots of T2D TWAS results for different frameworks. Top: scPrediXcan in β cells from the T2D dataset. Center: TWAS-pseudobulk in β cells from the T2D dataset. Bottom: TWAS-bulk in pancreas tissue from GTEx dataset. The red dashed lines are Bonferroni-corrected thresholds (*p* < 0.05/number of genes in the association study).

(B) QQ plot of TWAS *p* values in T2D between frameworks.

(C) QQ plot of TWAS *p* values in T2D of scPrediXcan against the null distribution (i.e., uniform distribution). Blue, silver-standard genes; orange, other genes; dashed lines, Bonferroni-corrected thresholds.

(D) Bar plot of TWAS −log10(p) of T2D silver-standard genes in different frameworks. Top: scPrediXcan in β cells from the T2D dataset. Center: TWAS-pseudobulk in β cells from the T2D dataset. Bottom: TWAS-bulk in pancreas tissue from the GTEx dataset.

(E) UpSet plot of scPrediXcan-nominated candidate causal genes for T2D in different cell types.

(F) Scatterplot of precision and recall of different gene prioritization methods for T2D causal gene nomination.

scPrediXcan has 24 Bonferroni-corrected significant genes ($p < 2.7 \times 10^{-6}$), whereas TWAS-pseudobulk and TWAS-bulk identify only 4 ($p < 2.8 \times 10^{-6}$) and 13 ($p < 8.5 \times 10^{-6}$) significant genes, respectively. Notably, both scPrediXcan and TWAS-pseudobulk recognize *IGF2BP2*; scPrediXcan and TWAS-bulk concurrently identify five silver-standard genes, underscoring scPrediXcan's higher sensitivity in detecting T2D-related genes.

To investigate the cell-type specificity of the association with T2D risk, we analyzed the scPrediXcan results for all 11 islet cell types in the T2D dataset (Figure 5E). We observed that, while most TWAS hits were common across different cell types—48 (9.3%) genes are shared by all and 392 (76.1%) appeared in at least two cell types—123 (23.9%) genes are unique to one cell type. To further examine the cell-type specificity of the 123 genes identified in only one cell type, we aggregated $p$ values across all remaining cell types using the ACAT method. After aggregation, two genes reached Bonferroni significance, while 118 genes were nominally significant ($p < 0.05$) in other cell types—these are referred to as cell-type-enriched genes. The remaining three genes (*CYB561*, *PISD*, and *RREB1*), referred to as cell-type–specific genes, did not reach nominal significance in any other cell type (Figure S2A). This suggests that associations involving cell-type-enriched genes may not be strictly cell type specific but, rather, enriched in the focal cell type.

Our cell-type- enriched results also highlight several genes previously proposed as candidate drivers of T2D that were missed by TWAS-bulk analyses conducted on pancreas tissue, illustrating the advantages of performing TWAS at the cell type level. For example, the gene *CASR*, which mediates white adipose tissue dysfunction to promote the development of obesity-induced T2D,[26] reached significance only in gamma cells ($p = 2.3 \times 10^{-6}$); *MSRA*, which can cause oxidative stress when downregulated, leading to obesity-induced T2D,[27] is identified only in activated stellate cells ($p = 1.6 \times 10^{-7}$), and *LPL*, associated with a lower risk of T2D when upregulated,[28] is found exclusively in quiescent stellate cells ($p = 7.6 \times 10^{-8}$). These findings highlight the potential of scPrediXcan to uncover nuanced, cell-type-enriched pathways involved in disease processes. The scPrediXcan T2D results for all cell types are provided in Tables S4–S14.

Finally, we benchmarked scPrediXcan, TWAS-pseudobulk, and TWAS-bulk against five other gene prioritization methods—Effector Index,[29] Polygenic Priority Score,[30] 3D chromatin,[31] CRISPR-screen,[32] and COLOC[33,34]—using the T2D silver-standard genes to evaluate each method's precision and recall (Figure 5F; STAR Methods). Among the eight gene prioritization methods, scPrediXcan demonstrates the second-highest recall at 0.439, surpassed only by the effector index method's 0.561. Although scPrediXcan ranks fourth in precision (0.109), it is noteworthy that precision scores are generally low across all computational methods (0.046–0.175), except for the CRISPR-screen method (0.40). This pattern suggests that PrediXcan's limited precision may stem from the silver-standard gene list not fully capturing the complex genetic landscape of T2D. The high recall rate of scPrediXcan highlights its robust ability to identify relevant genes, demonstrating its effectiveness despite the precision limitations of computational gene prioritization methods.

## scPrediXcan in SLE outperforms canonical TWAS methods by identifying promising candidate causal genes in more genomic loci

Next, we performed TWAS for SLE using scPrediXcan and benchmarked its performance against the two canonical TWAS methods (pseudobulk and bulk; STAR Methods). scPrediXcan identified a greater number of candidate causal genes and explained more GWAS loci for SLE than the TWAS-pseudobulk and TWAS-bulk frameworks. Representative results for transitional B cells (scPrediXcan and TWAS-pseudobulk) and whole blood (TWAS-bulk) are shown in Figure 6A.

To evaluate the effectiveness of our approach, we conducted a two-step comparison of the TWAS $p$ values. First, for genes predicted by all three models, we directly compared the $p$ values (Figure S5). For a fairer comparison, we imputed genes that were not predicted by the canonical frameworks with uniformly distributed $p$ values. This comparison shows that scPrediXcan significantly outperforms the other methods (Figure 6B).

Specifically, scPrediXcan identifies 129 candidate causal genes from 24 different LD blocks among 1,703 pre-defined LD blocks,[23] whereas TWAS-pseudobulk only identifies 11 candidate causal genes from 8 different LD blocks, and TWAS-bulk identifies 54 candidate causal genes from 14 different LD blocks. As expected, we noticed that both scPrediXcan and TWAS-bulk nominate many candidates at the human leukocyte antigen (HLA) region on chromosome 6. scPrediXcan also pinpoints significant genes on other chromosomes that the other frameworks miss. For example, only scPrediXcan identifies two genes on chromosome 16 that have been implicated in SLE pathogenesis: *PYCARD* (also named *ASC*) in CD4+ α-β T cells ($p = 7.8 \times 10^{-48}$) and *ITGAM* in CD14+ monocytes ($p = 4.3 \times 10^{-41}$).[35,36]

To investigate the cell-type specificity of the association with SLE risk, we analyzed the scPrediXcan results of 12 immune cell types. To simplify interpretation, we aggregated the results of the various subtypes of T cells using the ACAT method. Similar to our findings for T2D, we found that, while most TWAS hits are shared by different cell types, a few are cell type enriched, which we defined above as Bonferroni significant in one cell type and nominally significant in others (Figure 6C). Among 243 TWAS hits from the 12 immune cell types, 27 (11.1%) genes are shared in all cell types, 205 (84.3%) genes are shared in at least two cell types, and 38 (15.6%) genes yield significance in a single cell type. The full set of scPrediXcan SLE results is shown in Tables 15–43.

To further investigate the cell-type specificity of the 38 genes identified in only one cell type, we aggregated $p$ values across all remaining cell types using the ACAT method. Our analysis revealed that 18 of these genes were cell type enriched (i.e., nominally significant [$p < 0.05$]) in non-focal cell types. The remaining 20 genes were cell type specific, as the association did not reach nominal significance ($p > 0.05$) even after aggregation across all non-focal cell types (Figure S2B).

We identified several potential driver genes for SLE among the cell-type-specific and cell-type-enriched associations that were overlooked in the bulk TWAS analysis. One notable example is the complement factor B (*CFB*) gene identified by scPrediXcan as a cell-type-specific gene associated with SLE risk[37] in T cells ($p = 2.8 \times 10^{-8}$). Deficiencies in the classical complement

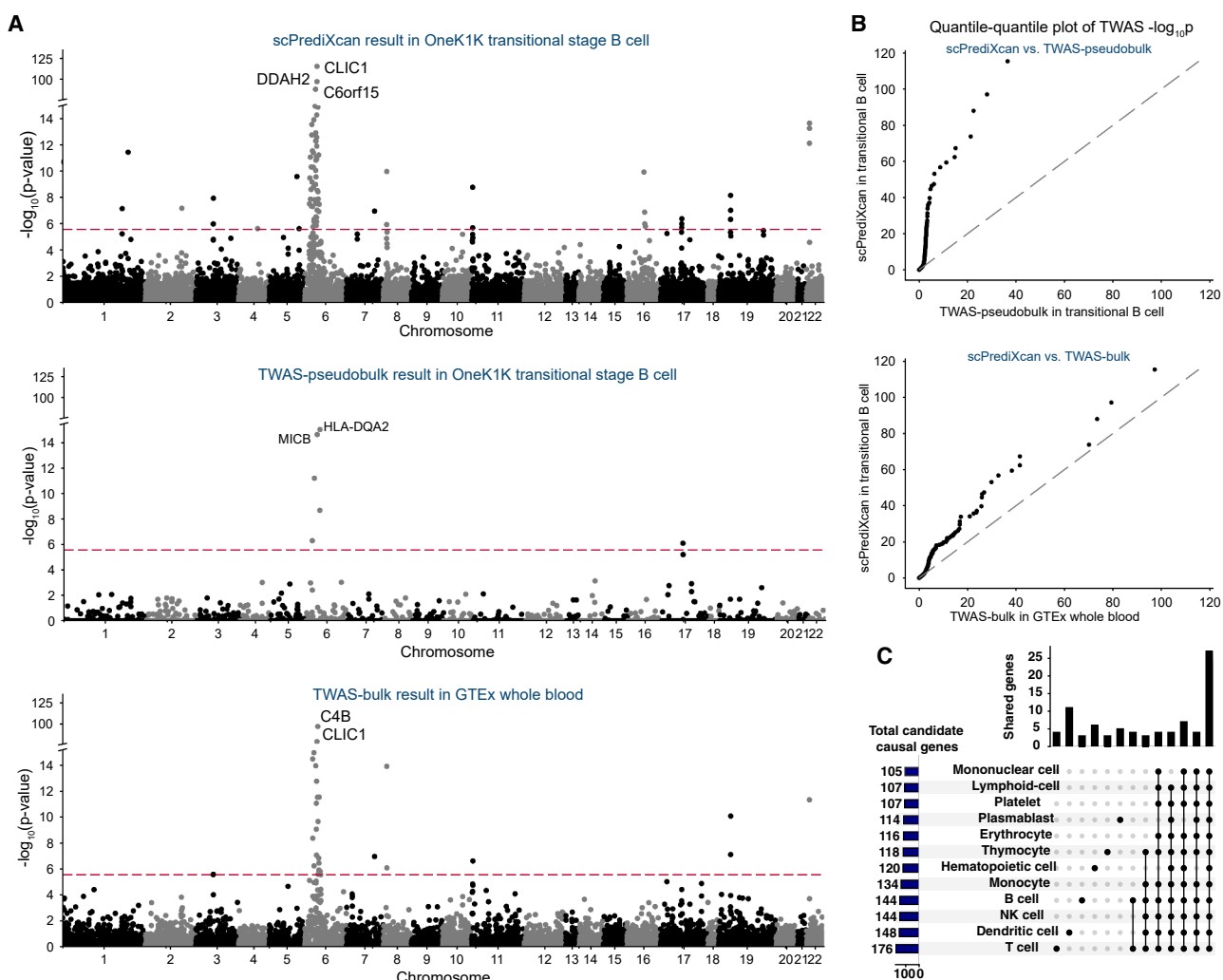

**Figure 6. scPrediXcan in SLE outperforms canonical TWAS methods**
(A) Manhattan plots for different SLE TWAS frameworks. Top: scPrediXcan in transitional B cells from the OneK1K dataset. Center: TWAS-pseudobulk in transitional B cell from the OneK1K dataset. Bottom: TWAS-bulk in whole blood from the GTEx dataset. The red dashed lines are Bonferroni-corrected thresholds ($p < 0.05$/number of genes in the association study).
(B) QQ plots of SLE TWAS $p$ values between frameworks.
(C) UpSet plot of scPrediXcan-nominated candidate causal genes for SLE in different cell types.

pathway significantly contribute to SLE predisposition, as its disruption impairs the clearance of apoptotic cells and initiates an autoimmune response through the recognition of cellular debris by autoantibodies. This leads to a loss of tolerance by antigen-presenting cells and subsequent activation of T cells.[38,39] Importantly, *CFB* is part of the alternate complement pathway. Genes in these pathways are crucial for cellular clearance and immunity, and their upregulation has been reported in other autoimmune diseases, like lupus nephritis,[40] suggesting potential therapeutic utility. Another example is *CXCR5*, identified in plasmablast cells ($p = 1.4 \times 10^{-6}$), which has been found to be differentially expressed in SLE patients compared to healthy controls.[41] Notably, *CXCR5* has been reported to be critically involved in the progression of SLE.[42] These examples showcase the ability of scPrediXcan to connect GWAS loci with genes known to have roles in SLE

and other diseases but that had been missed by canonical TWAS and other expression-based integrative approaches.

## DISCUSSION

In this study, we presented scPrediXcan, a framework designed to perform TWASs at the cell type level. By leveraging the epigenetic representation of DNA sequences learned by a pre-trained deep-learning model and integrating single-cell expression data, we trained cell-type-specific gene expression predictors. We applied scPrediXcan to both T2D and SLE, benchmarking our results against canonical TWAS frameworks under various training conditions. Our findings indicate that scPrediXcan not only identifies a larger number of candidate causal genes and explains more GWAS loci but also exhibits higher power in identifying

candidate causal genes, especially when tested against a curated set of silver-standard T2D genes and genes with prior evidence of involvement in SLE. Moreover, scPrediXcan demonstrated enhanced sensitivity in nominating candidate causal T2D genes compared to other gene prioritization methods, such as COLOC, polygenic priority score, and effector index, among others.

Three key factors contribute to the enhanced performance of scPrediXcan. First, scPrediXcan utilizes a cross-genome deep learning model for gene expression prediction using reads aggregated across individuals, effectively reducing the impact of data sparsity and leveraging insights from a pre-trained sequence-to-epigenomics model. This strategy enables the prediction of a broader array of genes with high accuracy. Second, unlike the canonical TWAS framework, which typically operates at the tissue level, scPrediXcan focuses on the cell type level. This granularity provides a more resolved context for identifying putative causal genes and captures those that might be overlooked at the tissue level. Third, scPrediXcan's lower sample size requirement allows for the utilization of patient data that are often more disease relevant, but less available, than data from healthy controls. This aspect of scPrediXcan is particularly advantageous, as it can reveal candidate disease drivers that may remain hidden in non-disease contexts.

In summary, scPrediXcan leverages single-cell RNA-seq data and GWAS summary statistics to perform TWASs at the cell type level, offering significant potential to identify putative causal genes in disease-relevant cell types. This capability advances our understanding of disease etiology and supports future experimental and clinical research.

To facilitate broad adoption, we make scPrediXcan and its integrated SNP-based linear predictors for 46 cell types publicly available at predictdb.org, providing a user-friendly tool for nominating context-specific causal genes. This resource can be used by end users without expertise or infrastructure to perform deep learning analysis.

### Limitations of the study

While scPrediXcan presents a robust framework for conducting TWAS at the cell type level, there are certain limitations to consider. First, ctPred uses Enformer as its input, and some of its predictions correlate negatively with observed expression levels due to challenges inherent to Enformer. This issue is not unique to Enformer but represents a common limitation across current sequence-based deep learning models that predict molecular phenotypes from reference genomes.[16,17] Hence, our current analysis focuses on the $p$ values of correlations and associations and does not consider the direction of correlation. However, we are still limited in our ability to discern whether disease risk is associated with up- or downregulation of a nominated gene, a crucial piece of information for drug target selection and development. We are actively working to refine this aspect, with improvements anticipated as enhanced pre-trained models become available. Second, Enformer primarily captures gene expression determinants in promoter regions while largely overlooking distal enhancers,[43] whose activities are often cell type dependent and important in regulation. Because ctPred relies on Enformer predictions as input, it inherits this limitation,

missing molecular information in some cell-type-specific enhancer regions, which may impact its predictive performance. Moreover, like most TWAS frameworks, scPrediXcan primarily focuses on *cis*-regulatory mechanisms and does not account for *trans* effects or other regulatory mechanisms, and it is susceptible to LD-induced errors. Third, our method ignores prediction error, which reduces the power of the association. Addressing this prediction uncertainty within the framework is essential for enhancing the power and accuracy of associations in future iterations of the method. Fourth, given that GWASs and other omics data are predominantly obtained from European samples, it is important to assess the performance of our methods in non-European populations. For applications involving these populations, using ancestry-matched *in silico* references is likely to provide the greatest benefit when working with GWAS summary statistics. We made our software and pipelines publicly accessible so that ancestry-specific reference expression data can be easily generated and used for training genetic predictors.

### RESOURCE AVAILABILITY

#### Lead contact
Requests for further information and resources should be directed to and will be fulfilled by the lead contact, Hae Kyung Im (haky@uchicago.edu).

#### Materials availability
This study did not generate new unique reagents.

#### Data and code availability
The code for scPrediXcan is available at https://github.com/hakyimlab/scPrediXcan (https://doi.org/10.5281/zenodo.15014568). Prediction models are available at https://predictdb.org (https://doi.org/10.5281/zenodo.15318900 and https://doi.org/10.5281/zenodo.14346661).

### ACKNOWLEDGMENTS

We thank BioRender.com for providing the platform used to generate the graphical abstract and Figure 1. This research used resources of the Argonne Leadership Computing Facility, a DOE Office of Science User Facility supported under contract DE-AC02-06CH11357. This work was completed in part with resources provided by the University of Chicago's Research Computing Center and Beagle3. We also acknowledge resources from the Center for Research Informatics, funded by the Biological Sciences Division at the University of Chicago, with additional funding provided by the Institute for Translational Medicine, CTSA grant 2U54TR002389-06 from the National Institutes of Health. Y.Z., M.C., and H.K.I. were partially funded by R01 GM126553 and R01 HG011883. Y.Z., S.S.-M., T.A., and H.K.I. were funded in part by R01AA029688. H.K.I. was funded in part byP30DK020595.

### AUTHOR CONTRIBUTIONS

Conceptualization, Y.Z, M.C., and H.K.I.; methodology, Y.Z., T.A., S.G., F.N., and and H.K.I.; software, Y.Z., T.A., S.G., F.N., and and H.K.I.; formal analysis, Y.Z. and H.K.I.; data curation, Y.Z. and F.N.; visualization, Y.Z. and S.S.-M.; resources, M.C., R.M., R.K., H.K., J.E.P., and H.K.I.; writing – original draft, Y.Z. and S.S.; writing – review & editing, Y.Z., S.S., S.S.-M., B.L., M.C., and H.K.I.; funding acquisition, M.C. and H.K.I.

### DECLARATION OF INTERESTS

The authors declare no competing interests.

## DECLARATION OF GENERATIVE AI AND AI-ASSISTED TECHNOLOGIES IN THE WRITING PROCESS

During the preparation of this work, the authors used ChatGPT (GPT-4-turbo) to improve the grammar and flow of the paragraphs. After using these tools, the authors reviewed and edited the content as needed and take full responsibility for the content of the publication.

## STAR★METHODS

Detailed methods are provided in the online version of this paper and include the following:

- KEY RESOURCES TABLE
- METHOD DETAILS
  - Data
  - Computational methods
  - Canonical TWAS framework
  - T2D-associated gene prioritization methods comparison

## SUPPLEMENTAL INFORMATION

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

## Article

CellPress

# STAR★METHODS

## KEY RESOURCES TABLE

| REAGENT or RESOURCE | SOURCE | IDENTIFIER |
|---|---|---|
| **Deposited data** | | |
| OneK1K single-cell RNA-seq data | Yazar et al.[13] | https://cellxgene.cziscience.com/collections/dde06e0f-ab3b-46be-96a2-a8082383c4a1 |
| Tabula Sapiens single-cell RNA-seq data | The Tabula Sapiens Consortium (The T2D single-cell RNA-seq data is available on request until the accession is live from the lead author of the other paper that generated the dataset) et al.[14] | https://figshare.com/articles/dataset/Tabula_Sapiens_v2/27921984 |
| T2D single-cell RNA-seq data | Kim et al. | GSE234313 (The T2D single-cell RNA-seq data is available on request until the accession is live from the lead author of the other paper that generated the dataset) |
| **Software and algorithms** | | |
| scPrediXcan | This paper | https://github.com/hakyimlab/scPrediXcan |
| PrediXcan | Gamazon et al.[1] | https://github.com/hakyimlab/PrediXcan |
| S-PrediXcan | Barbeira et al.[3] | https://github.com/hakyimlab.MetaXcan |
| Enformer | Avsec et al.[9] | https://github.com/google-deepmind/deepmind-research/tree/master/enformer |
| UTMOST | Hu et al.[24] | https://github.com/Joker-Jerome/UTMOST |
| SHAP | Lundberg et al.[44] | https://github.com/shap/shap |

## METHOD DETAILS

### Data

#### Single-cell transcriptomics data and genotype data

The raw cell by gene matrices of OneK1K single-cell transcriptomics data and genotype data were shared by Dr. Joseph Powell. The data was pre-processed by Dr. Powell's team described briefly as follows. The cell-droplets identified as doublets by both Demuxlet[45] and Scrublet[46] were removed. For each pool of cells captured, the distributions of total number of UMIs, number of genes, and percentage of mitochondrial gene expression were normalized using an Ordered Quantile Transformation. Then a generalized linear model with SCTransform[47] method was used to account for batch effects and to get the gene unique molecular identifier (UMI) count matrix. Cells were classified into the major immune populations in a supervised manner using the gene expression data by a digital single cell transcriptional profiling panel[48] as a reference. After the pre-processing steps done by Dr. Powell's team, for further quality control, we selected cells with 'nCount_RNA' less than 10000 to avoid potential doublets and multilets, and the percentage of mitochondria RNA less than 10%. Then, we used the filtered data with its original cell type annotations reference for the downstream analysis.

The T2D single-cell transcriptomics data and genotype data were shared by Dr. Rohit Kulkarni. The data was pre-processed by Dr. Rohit Kulkarni's team: the UMI count matrix was filtered by quality control requirements for cells to express at least 200 gene features and each gene feature to be present in at least three cells. Then doublets and triplets were removed using DoubletDecon.[49] Cells were manually annotated using a list of canonical markers from previous human islet single-cell RNA-seq studies. After the pre-processing steps done by Dr. Rohit Kulkarni's team, the processed data with its cell type annotations was used for the downstream analysis.

Tabula Sapiens single-cell transcriptomics data was downloaded online.[14] The data was pre-processed by the Tabula Sapiens Consortium. Briefly, cells that did not have at least 200 detected genes were removed, and then cells with fewer than 5000 counts and for droplet cells with fewer than 2500 UMIs were removed. DecontX[50] was used to filter out reads from ambient RNA. The dataset was re-filtered by excluding the mitochondrial encoded genes when removing cells that did not contain the minimum number of genes and/or minimum of counts/UMIs to get the gene-count matrix. Cells were classified into different cell types using annotation methods include random forest (RF),[51] support vector machine (SVM),[51] scANVI,[52] onClass,[53] and k nearest neighbors (kNN) after batch-correction using single-cell harmonization methods (scVI,[54] BBKNN,[55] Scanorama[56]). After the preprocessing steps done by Tabula Sapiens team, we used the filtered gene-count matrix with cell annotations for the downstream analysis.

### GWAS summary statistics

We used the multi-ancestry GWAS meta-analysis summary statistics[30] from the DIAGRAM (DIAbetes Genetics Replication and Meta-analysis) consortium. We lifted over SNP coordinates to hg38 coordinates using UCSCs liftover tool to map variants between genome builds.

We downloaded the GWAS summary statistics of SLE[35] from the GWAS Catalog.

### Computational methods
#### scPrediXcan framework

ctPred model training and prediction: *Training data preprocessing:* We used Enformer to predict the epigenomic features surrounding the transcription start site (TSS) of each gene. We fed Enformer with the required sequence length of 196,608 base pairs (bp) centered at the TSS and generated a 5,313 x 896 matrix representing the central 114,688 bp window. Within this matrix, each of the 896 bins contained one epigenomic feature for a 128-bp segment, encompassing a total of 5,313 distinct features.

To reduce the computational burden for the training of ctPred, we averaged the central four bins (447-450th) as the local regions of gene TSS into a linear 1 by 5,315 vector as the final epigenomic representation of the gene. The central four bins were determined empirically. With a total of 896 bins, bins 448 and 449 are central. We tested various numbers of bins for model training, ranging from 2 to 12, and found that using four bins yielded the best performance, although the differences were not substantial. We determined this method maintained the best prediction performance through empirical testing, The epigenomic representations of genes were used as the inputs for the ctPred model.

For each cell-type–specific gene-by-individual count matrix at pseudobulk level, we averaged the read counts of each gene across individuals, ranked the averaged counts for each gene, and converted the ranks into percentiles ranging from 0 to 1. These percentile values were used as the target outputs for the ctPred model (Figure S7).

The minimum number of cells per cell type used for ctPred training was 125, with a corresponding minimum total read count of 561,372 reads per cell type. Although increasing the number of cells and total read counts generally improves ctPred prediction performance (Figure S6), the chosen thresholds for cell numbers and read counts were sufficient to maintain reasonable prediction accuracy.

*Fully-connected neural network training:* The model ctPred is a four-layer multi-layer perceptron (MLP) that predicts cell-type–specific gene expression levels given their epigenomic representations. The input is the reference epigenomic representation of a gene and the output is the rank-based gene expression percentile value for a specific cell type. Distinct models were trained for each cell type. The whole ctPred model consists of a linear layer that maps the 5,313-dimensional epigenomic representation to a hidden layer of 64 dimensions, followed by ReLU, dropout, and other three identical hidden layers to map to the final predictions. We split the protein-coding genes into training (14,429), validation (2,812), and test (2,426) sets by different chromosomes to avoid data leakage (Figure 2A). The MSE (mean squared error) loss was used as the loss function of the model. To avoid overfitting, we used dropout layers (dropout rate = 0.05) and applied weight decay (L2 regularization parameter = $5 \times 10^{-4}$). The model was saturated after 50–80 epochs. Finally, for model evaluation, the Pearson correlation between the observations and the predictions across genes in the test set was calculated.

*Personalized gene expression prediction using ctPred*: The personalized gene expression prediction process involves two inference steps: 1) Epigenomic Representation Inference: Personal genome sequences centered at each gene's TSS are input into Enformer to generate personalized gene epigenomic representations. 2) Gene Expression Prediction: These personalized epigenomic representations are used by ctPred to obtain individual predictions of gene expression.

ctPred model linearization: We generated an in-silico cell-type–specific gene expression reference dataset by predicting gene expressions for 462 European individuals from the 1000 Genomes Project. Specifically, the DNA sequences centered at each gene's TSS for different individuals were input into Enformer to obtain personalized gene representations. These representations were then processed by ctPred to predict individual gene expressions, forming the in-silico reference dataset. Subsequently, we linearized ctPred into ℓ-ctPred by fitting the genotype data and in-silico gene expressions with an SNP-based elastic net model, following the standard PrediXcan pipeline.[1] For validation, we calculated the 10-fold cross-validated Spearman correlations between ctPred predictions and ℓ-ctPred predictions for all genes (Figure 4).

This linearization is trying to find a linear approximation to a complex nonlinear function. There is no overfitting in this case because we test our linear predictors in individuals not used for training the l-ctPred models.

Performing association test by S-PrediXcan: We used the GWAS summary statistics of T2D and SLE, as well as the ℓ-ctPred model for TWAS. We calculated the Z score introduced by the Summary-PrediXcan method[12] to evaluate the associations between genes and the trait. The Z score of gene–trait association is calculated:

$$Z_g = \sum_{l \in scPred} \omega_{g,l} \frac{\widehat{\sigma}_l}{\widehat{\sigma}_g} \frac{\widehat{\beta}_l}{se(\widehat{\beta}_l)}$$

Where $\omega_{g,l}$ is the effect of $SNP_l$ on the $gene_g$, $\widehat{\sigma}_l$ is the estimated variance of $SNP_l$, and $\widehat{\sigma}_g$ is the estimated variance of gene expressions of $gene_g$, $\widehat{\beta}_l$ is the estimated effect size of $SNP_l$ on the trait and $se(\widehat{\beta}_l)$ is the standard error of the effect size of $\widehat{\beta}_l$. The $\omega_{g,l}$ is from the ℓ-ctPred model and the other statistics are calculated from GWAS summary statistics. Moreover, for the same gene–trait pair, a two-tailed *p-value* can be calculated from the Z score using a normal distribution. All the scPrediXcan results for T2D and SLE in different cell types are attached in the supplementary tables.

### Canonical TWAS framework

The canonical TWAS frameworks serve as the benchmark against which we evaluate scPrediXcan's performance.

#### *TWAS-pseudobulk*

*Pseudobulk elastic net (PEN) training and prediction:* The observed gene expression data at the pseudobulk level (i.e., a gene-by-individual count matrix for each cell type) was pre-processed using the same method as for ctPred. We averaged the counts for each gene across individuals, ranked the genes based on their average, and converted these ranks into percentiles ranging from 0 to 1. For each gene in a given cell type, we trained an SNP-based elastic net model by fitting the rank-based gene expression and the SNP-dosages at the *cis*-regions, following the standard PrediXcan pipeline.

To compare the PEN model against ctPred for personalized prediction in the T2D dataset, we divided the 29 individuals into a training set of 20 and a test set of 9. For downstream TWAS analysis, we used all 29 individuals for model training. In the OneK1K dataset, for model comparison against ctPred for personalized prediction, we randomly selected 800 individuals as the training set and 100 individuals as the test set. For downstream TWAS analysis, we used all individuals for model training.

*Performing association test by S-PrediXcan:* We followed the same workflow as the scPrediXcan association test using the S-PrediXcan method introduced earlier.

#### *TWAS-bulk*

*SNP-based elastic net:* For T2D, we directly downloaded the SNP-based elastic net models trained by GTEx pancreas bulk gene expression and genotype data ($n = 305$). For SLE, we directly downloaded the SNP-based elastic net models trained by GTEx whole blood bulk gene expression and genotype data ($n = 670$).

*Performing association test by S-PrediXcan:* We followed the same workflow as the scPrediXcan association test using the S-PrediXcan method introduced earlier.

### T2D-associated gene prioritization methods comparison

For scPrediXcan and TWAS-pseudobulk frameworks, we applied ACAT *p-value* combination method[21] to integrate the TWAS results of all the 11 cell types from islet in T2D dataset and used Bonferroni-corrected $p$ values ($p < 2.5 \times 10^{-6}$ for scPrediXcan, $p < 1.4 \times 10^{-5}$ for TWAS-pseudobulk) as the threshold to obtain the final prioritized gene lists. For TWAS-bulk, we directly used Bonferroni-corrected *p-values* ($p < 8.5 \times 10^{-6}$) as the threshold to obtain the final prioritized gene list from GTEx pancreas tissue. For other methods, the prioritized gene lists were downloaded from the CMDKP database. The silver-standard T2D-associated gene list was used to calculate the precision and recall of all the methods. The precision was defined as the number of nominated silver-standard genes divided by the total number of nominated genes, and the recall was defined as the number of nominated silver-standard genes divided by the total number of silver-standard genes.

