## [Document S2. Transparent peer review records for Zhou et al. · Cell Genomics]

scPrediXcan integrates deep learning methods and single-cell data into a cell-type-specific transcriptome-wide association study framework

Author list: Yichao Zhou, Temidayo Adeluwa, Lisha Zhu, Sofia Salazar-Magaña, Sarah Sumner, Hyunki Kim, Saideep Gona, Festus Nyasimi, Rohit Kulkarni, Joseph Powell, Ravi Madduri, Boxiang Liu, Mengjie Chen, Hae Kyung Im

Summary

Initial submission: Received : Oct 14th 2024

Scientific editor: Judith Nicholson

First round of review: Number of reviewers: 3
Revision invited : November 8th 2024
Revision received : January 31st 2025

Second round of review: Number of reviewers: 3
Accepted : 14th March 2025

Data freely available: Yes

Code freely available: Yes

This transparent peer review record is not systematically proofread, type-set, or edited. Special characters, formatting, and equations may fail to render properly. Standard procedural text within the editor's letters has been deleted for the sake of brevity, but all official correspondence specific to the manuscript has been preserved.

Referees' reports, first round of review

Reviewer 1:

The manuscript proposed scPrediXcan, aiming to integrate deep learning approaches that predict epigenetic features from DNA sequences with the canonical TWAS framework. The prediction method, ctPred, is able to predict cell type-specific expression with high accuracy and captures complex gene regulatory grammar that linear models overlook. Real data analysis for type 2 diabetes and systemic lupus erythematosus showed the advantage of scPrediXcan. Overall, scPrediXcan is a promising framework that extend the TWAS field, given its ability in providing insights into the cellular specificity of TWAS hits. I have the following comments that the authors should address for further improvement.

Major concerns

1. In Step 2 of scPrediXcan framework, the predicted gene expression treated as fixed value, which ignored the uncertainty in the prediction. In addition, in step 3 of scPrediXcan, the authors also treat the derived weights as fixed, which also ignore the uncertainty in inferring the SNP effect on the gene expression. I wonder whether the failure to account for uncertainty has an impact on the results.
2. Accounting for horizontal pleiotropy is important in canonical TWAS framework. Is it possible to extend scPrediXcan to further account for the cell type-specific horizontal pleiotropy.
3. Given that the gene expression is predicted by epigenomic features in scPrediXcan, can the gene expression still be interpreted the same as that in canonical TWAS, i.e. genetically regulated expression? I wonder whether we need to interpret is as gene expression regulated by both epigenomic information and SNPs?
4. In Step2 of scPrediXcan, the gene expression is predicted by individual epigenomic features, while the epigenomic features was obtained from individual

DNA sequences. Thus, the predicted gene expression conceptually depends on the individual DNA sequences, while the authors treated these gene expression as fixed and further re-linked them with the DNA sequences. That is, it seems that scPrediXcan uses the same DNA sequences from the same individuals two times, I wonder whether there are some statistical issues, e.g overfitting.

5. scPrediXcan finally uses the weight obtained from l-ctPred. I suggested the authors to compare the performance between l-ctPred and PEN.

6. The authors only compared scPrediXcan with one TWAS-bulk method. i.e. S-PrediXcan, I wonder what would happen if comparing scPrediXcan with other TWAS-bulk methods, such as HMAT, PMR-Egger, UTMOST.

Minor concerns

1. In Fig1, are the reference human genome in Step1 and the individual DNA sequence in Step2 required from the same ancestry?

2. In the processing of Tabula Sapiens dataset, the authors mention that a subset was selected, what is the selection criterion?

3. Is data12 in line 512 a typo?

4. Is the legend of Figure 4 correct?

5. Is π in the figure caption of Supplementary fig. 4 consistent with π in the manuscript?

Reviewer 2:

For the full text:

1. Abbreviations in the whole text: for example, TWAS, GWAS, T2D, SLE, TSS. The full name and abbreviation should be written when it appears for the first time, and only the abbreviation should be written when it appears again.

2. The p in the p -value in the whole text should be italicized.

3. Since this paper only uses part of the data from the Tabula Sapiens dataset, please consider whether to indicate that it is a Tabula Sapiens subset every time the dataset appears so that readers are not confused about the data details.

For the specific details:

4. From line 48 to line 57, the content is repeated. Please sort it out again.
5. In line 119, 'we analyzed a subset of the Tabula Sapiens dataset comprising 6 cell types from 15 individuals.' What is the basis for selecting 6 cell types from 15 individuals?
6. In line 151, 'aggregated across individuals (Supplementary Figures 7a, e, and i).' Please check whether the image citation is correct.
7. In line 189, what is the definition of π_1 ?
8. In line 231, cell type-specific is cell-type-specific? Similar problems also appear in line 837.
9. In line 232, in Fig. 3, please check whether the legends of Figures d and e are correct.
10. In line 235, predicted-gene is predicted gene?
11. In line 338, please check whether 121 is 123.
12. In line 497, RNAseq is RNA-seq?
13. In line 516, explanation of UMIs?
14. In lines 558 and 560, the full name of TSS should be written first when it appears for the first time. Write the abbreviation when it appears again.
15. In line 559, 'We fed Enformer with a sequence length of 196,608 base pairs (bp)'. How was 196,608 determined?
16. In line 565, how is 'four bins (447-450th)' determined?
17. In lines 659 and 661, 'Bonferroni-corrected p values ($p < 2.5 \times 10^{-6}$ for scPrediXcan, $p < 1.4 \times 10^{-5}$ for TWAS-pseudobulk), Bonferroni-corrected p-values ($p < 8.5 \times 10^{-6}$)' What are the criteria for determining these specific Bonferroni-corrected p values?
18. In line 888, Is the y-axis of Figure d wrong?
19. In line 904, there is an extra vertical line after scRNA-seq.
20. In line 277, 'whereas ℓ -ctPred relies on in-silico ctPred-predicted gene expression data.' In this study, can in-silico data completely replace real-world data? Is it sufficient to use only Spearman correlations (line 610) to evaluate ctPred predictions and ℓ -ctPred predictions?

Reviewer 3:

The authors of this manuscript present and apply a framework, scPrediXcan, that performs transcriptome-wide association studies (TWAS) at the cell type level. They do this by making use of a popular pre-trained deep-learning based tool,

Enformer, to generate informative regulatory features from genomic sequence as input to their deep learning model (MLP), ctPred, which is trained to predict cell type specific gene expression ranks/percentiles. The authors report that applying this framework, as is, to perform TWAS at scale (e.g., predicting gene expression ranks for 500 individuals across 20,000 genes) would be computationally unfeasible, requiring ~2,700 GPU-hours. Thus, to perform TWAS studies, they generate a linearized version of their ctPred model, l-ctPred, by performing elastic net linear regression using Enformer output as their input features and the gene expression ranks generated from ctPred. They apply their framework to type 2 diabetes (T2D) and systemic lupus erythematosus (SLE) and demonstrate that it outperforms canonical TWAS approaches in identifying more candidate genes and effectively explaining more genome-wide association studies (GWAS) loci while yielding cell type specific insights. The authors provide a useful tool to the community of investigators interested in performing TWAS study, and their analysis of T2D and SLE datasets is extensive. The manuscript could be improved by making more clear input-output relationships with respect to each of the models in the main text as well as providing brief summaries of Enformer's specific outputs and its limitations. It would be helpful to explore Enformer's limitations in the context of this study to see, for example, if they explain their incorrect prediction trends (i.e., anticorrelation between observed and predicted gene expression percentiles for KCNJ13 and DPH1 shown in Fig. 3a).

Major comments:

1. Given that a major selling point of the scPredXcan framework is its ability to perform cell type specific TWAS and its reliance on Enformer to derive epigenomic information from DNA sequence, brief summaries of how Enformer works, its explicit output features and its limitations would help a more general audience who are not familiar with Enformer. It would also help provide some context for scPredXcan's strengths and weaknesses. Below are more detailed suggestions.

- 1a. When introducing Enformer in the main text, I would summarize that Enformer was trained and tested using human and mouse genomic sequence as input and outputs, for human, 2,131 transcription factor (TF) chromatin immunoprecipitation and sequencing (ChIP-seq), 1,860 histone modification ChIP-seq, 684 DNase-seq or ATAC-seq, and 638 CAGE profiles derived from a wide

range of different cell lines (i.e., a total 5,313 output features). Focusing on human, for any given genomic locus, it takes 196,608bp of DNA as input and learns/predicts a 5,313 x 896 output matrix of the all the 5,313 profiles whose values are aggregated into 128bp bins.

1b. To anyone with a reasonable understanding of epigenetics/epigenomics, an extreme limitation and contradiction will immediately come to mind. For the same genomic sequence of any given individual, Enformer will output the same set of 5,313 x 896 matrices/profiles across the genome regardless of the cell type. Thus, while Enformer has effectively memorized a wide range of epigenomic profiles across cell types, its output will be cell type independent given a specific genome! This is a severe limitation of Enformer and should be highlighted. Running with this line of argument further, one could ask, what genes epigenomic profiles might be relatively constant across cell types? One simple answer is housekeeping genes. What is well known in the epigenetics field is that housekeeping genes tend to be promoter regulated while genes that are highly cell type specific tend to be regulated by enhancers. Indeed, papers that have further evaluated Enformer and methods like it find that it tends to capture promoter regulation while ignoring enhancers, which are major drivers of cell type specific regulation (see for example the following paper by Karollus et al.: "Current sequence-based models capture gene expression determinants in promoters but mostly ignore distal enhancers" *Genome Biology* 24 (2023)). This paper and others along these lines should be cited while putting the limitations of Enformer in proper context including how it may limit scPredXcan. Specifically, scPredXcan is using input features that are not themselves cell type specific and only weakly encode cell type specific information. The authors can summarize this limitation much more briefly than the argument I provide above.

1c. It would be very informative and helpful to readers of this paper and potential users of scPredXcan to understand how the limitation of Enformer affects scPredXcan. Does this limitation explain ctPred's incorrect prediction trends (i.e., anticorrelation between observed and predicted gene expression percentiles for KCN13 and DPH1 shown in Fig. 3a)? This could occur because genomic sequence promoter signals are effectively weakening for KCN13 and DPH1 but they are cell type specific genes controlled more by their respective enhancers? One kind of analysis that would be useful would be to classify genes that the authors identify and miss (that other approaches that they compare their method to identify)

according to the degree that they are cell type specific. Another interesting way to evaluate how scPred is effectively working is to perform input feature importance analysis. This can be done with DeepLift or by calculating Shapley values. Do relevant cell type specific input features (e.g., CAGE or epigenetic features from T-cells) explain model predictions (i.e., gene expression ranks for T-cells)? This analysis would only have to be performed for one of ctPred's models to be informative.

2. The input-output relationships of ctPred and linear-ctPred should be made more clear in the main text and Figure 1. Specifically, I would make more clear in the main text that ctPred takes features derived from Enformer as input and is trained to predict cell type specific gene expression ranks/percentiles as output. Along these lines, I would remove the red arrows and "MLP model training" in the top panel (Step 1) of Fig. 1 and add an arrow to something (text or graphic) indicating gene expression ranks/percentiles as output to ctPred. If the authors want, I suggest adding the MLP model training in red under the ctPred graphic. Similarly, in the second panel (Step 2) of Fig. 1, I would remove ctPred and the violin plots and have the Enformer features going directly to linear-ctPred graphic and add an arrow to the output of linear-ctPred which points to something indicating that its output is ctPred derived gene expression ranks/percentiles. Again, if the authors wish, they can add in red below the linear-ctPred graphic the text "Elastic net training".

Minor comments:

1. The authors should remove text where they frame their method as a form of "transfer learning". To my understanding, transfer learning entails starting with a pretrained model whose weights have been frozen as a starting point for further model training/fine tuning. This is opposed to starting with random weights. This approach is particularly useful when there is abundant related/relevant data for a specific task for which there is limited data. The authors simply use the output of Enformer, not its model architecture which they further fine-tune with their own training data. If the authors disagree, they should cite a paper that defines the approach they take as transfer learning as opposed to what I'm detailing above. I believe keeping the framing of their method as "transfer learning" will confuse machine learning experts.

2. In Fig. 3 legend, "GReX of in different..." should be "GReX in different...".
 3. Should the reference in line 403 be to Fig. 6b?
 4. In line 572, I would remove "across all individuals" as this is confusing because you already averaged them across individuals.
 5. In the Methods section, I would replace the inaccurate reference to the output of ctPred as being "gene expression level" with "gene expression ranks/percentiles" or "gene expression percentiles". A good example of this is in line 583, but it occurs in other places. Supplementary fig. 2 should also be corrected along these lines: replacing "values" with "percentiles". I would also consider renaming the axes of plots where "Observed expressions" and "Predicted expressions" is used and add "percentiles". This is the case throughout the manuscript where these scatter plots are presented.
 6. In line 623, should you add "using a normal distribution" after "from the z-score"? How did you calculate a p-value from a z-score? Presumably via a normal distribution.
 7. In Supplementary fig. 7d, the y-axis is mislabeled and should be the same as panel b.
-

Authors' response to the first round of review

Reviewers' Comments:

Reviewer #1: The manuscript proposed scPrediXcan, aiming to integrate deep learning approaches that predict epigenetic features from DNA sequences with the canonical TWAS framework. The prediction method, ctPred, is able to predict cell type-specific expression with high accuracy and captures complex gene regulatory grammar that linear models overlook. Real data analysis for type 2 diabetes and systemic lupus erythematosus showed the advantage of scPrediXcan. Overall, scPrediXcan is a promising framework that extends the TWAS field, given its ability in providing insights into the cellular specificity of TWAS hits. I have the following comments that the authors should address for further improvement.

Major concerns

1. In Step 2 of scPrediXcan framework, the predicted gene expression is treated as fixed value, which ignores the uncertainty in the prediction. In addition, in step 3 of scPrediXcan, the authors also treat the derived weights as fixed, which also ignores the uncertainty in inferring the SNP effect on the gene expression. I wonder whether the failure to account for uncertainty has an impact on the results.

Response:

In Liang et al. (2024), we demonstrated that uncertainty in prediction does not compromise the validity (type I error) of TWAS associations, provided the error is independent of the target trait. However, ignoring this uncertainty may reduce statistical power. While accounting for prediction uncertainty is currently infeasible within the scPrediXcan framework, it represents a promising direction for future development.

We have added the following paragraph in the discussion section (lines 519-522):

“Third, our method ignores prediction error, which reduces the power of the association. Addressing this prediction uncertainty and incorporating it into the framework is essential for enhancing the power and accuracy of associations in future iterations of the method.”

2. Accounting for horizontal pleiotropy is important in the canonical TWAS framework. Is it possible to extend scPrediXcan to further account for the cell type-specific horizontal pleiotropy.

Response:

This topic is an active research direction for our team. In our paper, Liang et al. (2024), we present a method to address one form of horizontal pleiotropy (driven by the underlying polygenicity of the trait) that is compatible with all canonical TWAS frameworks.

The scPrediXcan framework is also fully compatible with our proposed method for adjusting polygenicity-driven horizontal pleiotropy. Here, we provide an example of applying this approach to scPrediXcan's T2D results in Beta cells. As suggested, accounting for cell-type-specific horizontal pleiotropy represents a promising future enhancement to the current scPrediXcan framework.

See the unpublished figures below:

3. Given that the gene expression is predicted by epigenomic features in scPrediXcan, can the gene expression still be interpreted the same as that in canonical TWAS, i.e. genetically regulated expression? I wonder whether we need to interpret it as gene expression regulated by both epigenomic information and SNPs?

Response: This would be a concern if we were using observed epigenetic features as the basis for predicting gene expression. However, the epigenetic features that are used to predict gene expression in ctPred/scPrediXcan are generated *in silico* by Enformer using a reference genome. Thus, we can safely assume that the epigenetic features we use to train ctPred—and by extension, the expression levels it predicts—are genetically determined.

We have added the following to the manuscript (Results, lines 92-95) to clarify this point: "*We used Enformer as a feature extractor: we used the genetically determined epigenomic features from Enformer to train ctPred—a lightweight, four-layer multi-layer perceptron (MLP) that generates single-cell expression data as output.*"

4. In Step2 of scPrediXcan, the gene expression is predicted by individual epigenomic features, while the epigenomic features was obtained from individual DNA sequences. Thus, the predicted gene expression conceptually depends on the individual DNA sequences, while the authors treated these gene expressions as fixed and further re-linked them with the DNA sequences. That is, it seems that scPrediXcan uses the same DNA sequences from the same individuals two times. I wonder whether there are some statistical issues, e.g. overfitting.

Response: The second step is trying to approximate the nonlinear prediction of gene expression from DNA sequence into a linear function of SNPs. This is akin to using a linear approximation to a complex nonlinear function. There is no overfitting in this case because we test our linear predictors in individuals not used for training the I-ctPred models.

We clarified this in the methods section, lines 644–646: “This linearization is trying to find a linear approximation to a complex nonlinear function. There is no overfitting in this case because we test our linear predictors in individuals not used for training the ℓ -ctPred models.”

5. scPrediXcan finally uses the weight obtained from ℓ -ctPred. I suggested the authors compare the performance between ℓ -ctPred and PEN.

Response: Thank you for the suggestion. We have included this comparison as supplementary figure 9 a–d, where we show that ℓ -ctPred also outperforms PEN.

We have added the following text to the main text of manuscript (Results, lines 278–280): “We also compared the performance of ℓ -ctPred to PEN in T2D and OneK1K datasets; like ctPred, ℓ -ctPred outperformed PEN across cell types. (Supplementary Fig 9).”

See the figure below:

Supplementary fig. 9: Comparison between ℓ -ctPred and PEN in predicting gene expression across individuals in different cell types.

Supplementary Fig. 9 a) Scatter plot of m_1 values for ℓ -ctPred and PEN in predicting gene expression across individuals for T2D cell types. Each dot is a cell type. **b)** Bar plot of m_1 values for ℓ -ctPred and PEN in predicting gene expression across individuals for T2D cell types. **c)** Scatter plot of m_1 values for ℓ -ctPred and PEN in predicting gene expression across individuals for OneK1K cell types. Each dot is a cell type. **d)** Bar plot of m_1 values for ℓ -ctPred and PEN in predicting gene expression across individuals for OneK1K cell types.

6. The authors only compared scPrediXcan with one TWAS-bulk method. i.e. S-PrediXcan, I wonder what would happen if comparing scPrediXcan with other TWAS-bulk methods, such as HMAT, PMR-Egger, UTMOST.

Response:

It is possible that other population-based methods may improve prediction or association results compared to S-PrediXcan, but our experience has been that improvements tend to be incremental. In comparison, ctPred shows an order of magnitude increase in the number of well-predicted genes (figure 3 d–e and supplementary table 2). This dramatic difference in performance is due to ctPred's training on orthogonal data obtained through Enformer. In response to the reviewer's request, we have added a comparison with UTMOST to illustrate this point in supplementary figure 11. We used the single tissue method in UTMOST. Our attempts to run the multi-tissue version did not work even with the help of the authors of UTMOST.

We added the following text to the main manuscript (lines 331–337) and added the following supplementary figure:

"To further evaluate scPrediXcan's performance compared to other canonical TWAS methods based on bulk data, we compared its results in beta cells to those of UTMOST, another canonical TWAS method applied to pancreatic tissue (Supplementary Fig. 11). scPrediXcan identified slightly more candidate genes (222 vs. 221) but achieved higher precision (0.109 vs. 0.082) and a significantly higher recall/power (0.439 vs. 0.153) in identifying T2D 'silver-standard' genes (described below). These results verify that scPrediXcan outperforms canonical TWAS methods in detecting functionally disease-relevant genes."

Supplementary fig. 11: scPrediXcan in T2D outperforms TWAS method UTMOST.
Supplementary Fig. 11 a) Manhattan plots of T2D TWAS results for different frameworks. Top: scPrediXcan in beta cell from T2D dataset. Bottom: UTMOST in pancreas tissue from GTEx dataset. The red dashed lines are Bonferroni-corrected thresholds ($p < 0.05/\text{number of genes in the association study}$). **b)** QQ-plot of TWAS p -values in T2D between frameworks. Shared genes: overlapped genes in both frameworks. All genes: all the protein-coding genes. Considering that ℓ -ctPred achieves convergence for significantly more genes than the SNP-based models used in the other two frameworks, we used a uniform distribution of p -values to represent genes absent in the canonical approaches, ensuring a comprehensive comparison. **c)** Scatter plot of precision and recall between scPrediXcan and UTMOST for T2D causal gene nomination.

Minor concerns

1. In Fig1, are the reference human genome in Step1 and the individual DNA sequence in Step2 required from the same ancestry?

Response: The method works without the need to match the ancestries of the references. For the training of Enformer, we expect that matching the ancestry of the reference genome and the epigenetic features would improve its performance and downstream the performance of scPrediXcan. Even better would be matching

the individual DNA with the epigenetic features, but despite efforts in the field, this has not been possible in part due to resource constraints.

Because of the use of GWAS summary statistics, it is likely to be more critical to match the ancestry of the GWAS and the in-silico reference expression. In this paper we have used European samples from 1000 genomes to match the majority of GWAS samples to date. As more diverse GWAS become available, ancestry-specific models can be trained using our user-friendly and publicly available software.

We have added the following to the discussion section (lines 522-527):

"Fourth, given that GWAS and other omics data are predominantly obtained from European samples, it is important to assess the performance of our methods in non-European populations. For applications involving these populations, using ancestry-matched in-silico references is likely to provide the greatest benefit when working with GWAS summary statistics. We have made our software and pipelines publicly accessible so that ancestry-specific reference expressions can be easily generated and used for training genetic predictors."

2. In the processing of Tabula Sapiens dataset, the authors mention that a subset was selected, what is the selection criterion?

Response: We no longer use a subset. We trained ctPred for all Tabula Sapiens cell types.

The text describing the analysis of datasets (lines 133-136) now reads: *"Further testing the model's robustness, we validated ctPred on 11 cell types from the T2D dataset and 178 cell types from the Tabula Sapiens subset (Supplementary table 46); we show the results from the T2D dataset and six representative cell types from Tabula Sapiens in Fig. 2c."*

3. Is data12 in line 512 a typo?

Response: Yes, this should be formatted as a citation. Thanks for pointing this out; we have corrected the error.

4. Is the legend of Figure 4 correct?

Response: No, the colors of two datasets should be interchanged. Thanks for pointing this out; we have corrected the error.

5. Is pi in the figure caption of Supplementary fig. 4 consistent with π in the manuscript?

Response: Yes, now we use π in both parts.

Reviewer #2: For the full text:

1. Abbreviations in the whole text: for example, TWAS, GWAS, T2D, SLE, TSS. The full name and abbreviation should be written when it appears for the first time, and only the abbreviation should be written when it appears again.

Response: We fixed the issue, thanks for the suggestion.

2. The p in the p-value in the whole text should be italicized.

Response: We fixed the issue, thanks for the suggestion.

3. Since this paper only uses part of the data from the Tabula Sapiens dataset, please consider whether to indicate that it is a Tabula Sapiens subset every time the dataset appears so that readers are not confused about the data details.

Response: We no longer use a subset. We trained ctPred for all Tabula Sapiens cell types.

The text describing the analysis of datasets (lines 133-136) now reads: "*Further testing the model's robustness, we validated ctPred on 11 cell types from the T2D dataset and 178 cell types from the Tabula Sapiens subset (Supplementary table 46); we show the results from the T2D dataset and six representative cell types from Tabula Sapiens in Fig. 2c.*"

4. From line 48 to line 57, the content is repeated. Please sort it out again.

Response: Thank you for pointing this out. We have deleted the repeated content now.

5. In line 119, 'we analyzed a subset of the Tabula Sapiens dataset comprising 6 cell types from 15 individuals.' What is the basis for selecting 6 cell types from 15 individuals?

Response: As mentioned above, we have trained all cell types in Tabula Sapiens. The 15 individuals are all the ones included in Tabula Sapiens.

6. In line 151, 'aggregated across individuals (Supplementary Figures 7a, e, and i).' Please check whether the image citation is correct.

Response: We fixed the issue, thanks for the suggestion.

7. In line 189, what is the definition of pi₁?

Response: We added the explanations, thanks for the suggestion.

8. In line 231, cell type-specific is cell-type-specific? Similar problems also appear in line 837.

Response: We fixed the issue, thanks for pointing it out.

9. In line 232, in Fig. 3, please check whether the legends of Figures d and e are correct.

Response: For clarity, we have revised them as follows:

"d) Bar plot of m1 values (i.e., number of true-positive genes) for ctPred and PEN in different cell types when compared against observed expression percentiles. e) Bar plot of m1 values (i.e., number of true positive genes) ctPred and PEN in different cell types when compared against GTEx GReX."

10. In line 235, predicted-gene is predicted gene?

Response: We fixed the issue, thanks for pointing it out.

11. In line 338, please check whether 121 is 123.

Response: Yes, that's a typo, and we fixed the issue, thanks for pointing it out.

12. In line 497, RNAseq is RNA-seq?

Response: Yes, now we use RNA-seq uniformly, thanks for pointing it out.

13. In line 516, explanation of UMIs?

Response: We added the full name as 'unique molecular identifier', thanks for pointing it out.

14. In lines 558 and 560, the full name of TSS should be written first when it appears for the first time. Write the abbreviation when it appears again.

Response: We added the full name as 'transcription start site', thanks for pointing it out.

15. In line 559, 'We fed Enformer with a sequence length of 196,608 base pairs (bp)'. How was 196,608 determined?

Response: The input length is defined by the Enformer architecture. The design of Enformer requires an input sequence length of 196,608 bp. We have clarified this in the text (line 589): "We fed Enformer with the required sequence length of 196,608 base pairs (bp)." For your convenience, we quote the Methods section of Avsec, et al. below:

"For the mouse genome, each example contains 308 TF ChIP-seq, 750 histone modification ChIP-seq, 228 DNase-seq or ATAC-seq, and 357 CAGE tracks (total 1,643, Supplementary Table 3). We modified the Basenji2 dataset by extending the input sequence to 196,608 bp from the original 131,072 bp using the hg38 reference genome." <https://www.nature.com/articles/s41592-021-01252-x#Sec8>

16. In line 565, how is 'four bins (447-450th)' determined?

Response: This was determined empirically. With a total of 896 bins, bins 448 and 449 are central. We tested various numbers of bins for model training, ranging from 2 to 12, and found that using four bins yielded the best performance, although the differences were not substantial.

We clarified this in the method section (lines 596–599), "*The central four bins were determined empirically. With a total of 896 bins, bins 448 and 449 are central. We tested various numbers of bins for model training, ranging from 2 to 12, and found that using four bins yielded the best performance, although the differences were not substantial.*"

17. In lines 659 and 661, 'Bonferroni-corrected p values ($p < 2.5 \times 10^{-6}$ for scPrediXcan, $p < 1.4 \times 10^{-5}$ for TWAS-pseudobulk), Bonferroni-corrected p-values ($p < 8.5 \times 10^{-6}$)' What are the criteria for determining these specific Bonferroni-corrected p values?

Response: All these p-values are calculated using $0.05/(\text{number of genes tested})$. Since the numbers of genes tested are different in scPrediXcan, TWAS-pseudobulk and TWAS-bulk methods, the thresholds are different.

18. In line 888, Is the y-axis of Figure d wrong?

Response: Good catch, thanks for pointing out! It should be 'across individuals', and we've fixed it.

19. In line 904, there is an extra vertical line after scRNA-seq.

Response: Good catch, thanks for pointing out! We've fixed it.

20. In line 277, 'whereas ℓ -ctPred relies on in-silico ctPred-predicted gene expression data.' In this study, can in-silico data completely replace real-world data? Is it sufficient to use only Spearman correlations (line 610) to evaluate ctPred predictions and ℓ -ctPred predictions?

Response: In this part of work, our goal was to approximate the ctPred predictions using linear predictors (ℓ -ctPred) to ensure the computational feasibility of the entire TWAS framework, not to replace real-world data with in-silico data. We demonstrated that this approximation was successful, as evidenced by the high Spearman correlations between the ctPred predictions and the ℓ -ctPred predictions in figure 4. Additionally, we showed in Figure 3 that ctPred outperformed pseudobulk elastic-net (PEN) when predicting across individuals. Furthermore, we now include an analysis comparing ℓ -ctPred to PEN performance when predicting across individuals in Supplementary Figure 9. As expected, ℓ -ctPred also outperformed PEN, showing higher numbers of m1 values across different cell types.

Reviewer #3: The authors of this manuscript present and apply a framework, scPrediXcan, that performs transcriptome-wide association studies (TWAS) at the cell type level. They do this by making use of a popular pre-trained deep-learning based tool, Enformer, to generate informative regulatory features from genomic sequence as input to their deep learning model (MLP), ctPred, which is trained to predict cell type specific gene expression ranks/percentiles. The authors report that applying this framework, as is, to perform TWAS at scale (e.g., predicting gene expression ranks for 500 individuals across 20,000 genes) would be computationally unfeasible, requiring ~2,700 GPU-hours. Thus, to perform TWAS studies, they generate a linearized version of their ctPred model, ℓ -ctPred, by performing elastic net linear regression using Enformer output as their input features and the gene expression ranks generated from ctPred. They apply their framework to type 2 diabetes (T2D) and systemic lupus erythematosus (SLE) and demonstrate that it outperforms canonical TWAS approaches in identifying more candidate genes and effectively explaining more genome-wide association studies (GWAS) loci while yielding cell type specific insights. The authors provide a useful tool to the community of investigators interested in performing TWAS study, and their analysis of T2D and SLE datasets is extensive.

The manuscript could be improved by making more clear input-output relationships with respect to each of the models in the main text as well as providing brief summaries of Enformer's specific outputs and its limitations.

It would be helpful to explore Enformer's limitations in the context of this study to see, for example, if they explain their incorrect prediction trends (i.e., anticorrelation between observed and predicted gene expression percentiles for KCNJ13 and DPH1 shown in Fig. 3a).

Major comments:

1. Given that a major selling point of the scPredXcan framework is its ability to perform cell type specific TWAS and its reliance on Enformer to derive epigenomic information from DNA sequence, brief summaries of how Enformer works, its explicit output features and its limitations would help a more general audience who are not familiar with Enformer. It would also help provide some context for scPredXcan's strengths and weaknesses. Below are more detailed suggestions.

1a. When introducing Enformer in the main text, I would summarize that Enformer was trained and tested using human and mouse genomic sequence as input and outputs, for human, 2,131 transcription factor (TF) chromatin immunoprecipitation and sequencing (ChIP-seq), 1,860 histone modification ChIP-seq, 684 DNase-seq or ATAC-seq, and 638 CAGE profiles derived from a wide range of different cell lines (i.e., a total 5,313 output features). Focusing on human, for any given genomic locus, it takes 196,608bp of DNA as input and learns/predicts a 5,313 x 896 output matrix of the all the 5,313 profiles whose values are aggregated into 128bp bins.

Response: Thanks for the suggestions, we added a summary of Enformer in the main text under 'Overview of scPredXcan framework' (lines 86–92).

"Enformer was trained and tested on human and mouse genomic sequences. When used for prediction in humans, Enformer takes a 196,608bp sequence of DNA as input (which we centered at each gene's transcription start site) and predicts a 5,313x896 output matrix. The 5,313 epigenomic features—comprising transcription factors, chromatin immunoprecipitation and sequencing, histone modifications, DNase-sequencing or assay for transposase-accessible chromatin using sequencing, and cap analysis of gene expression profiles—are aggregated into 128 bp bins."

1b. To anyone with a reasonable understanding of epigenetics/epigenomics, an extreme limitation and contradiction will immediately come to mind. For the same genomic sequence of any given individual, Enformer will output the same set of 5,313 x 896 matrices/profiles across the genome regardless of the cell type. Thus, while Enformer has effectively memorized a wide range of epigenomic profiles across cell types, its output will be cell type independent given a specific genome! This is a severe limitation of Enformer and should be highlighted. Running with this line of argument further, one could ask, what genes epigenomic profiles might be relatively constant across cell types? One simple answer is housekeeping genes. What is well known in the epigenetics field is that house keeping genes tend to be promoter regulated while genes that are highly cell type specific tend to be regulated by enhancers. Indeed, papers that have further evaluated Enformer and methods like it find that it tends to capture promoter regulation while ignoring enhancers, which are major drivers of cell type specific regulation (see for example the following paper by Karollus et al.: "Current sequence-based models capture gene expression determinants in promoters but mostly ignore distal enhancers" *Genome Biology* 24 (2023)). This paper and others along these lines should be cited while putting the limitations of Enformer in proper context including how it may limit scPredXcan. Specifically, scPredXcan is using input features that are not themselves cell type specific and only weakly encode cell type specific information. The authors can summarize this limitation much more briefly than the argument I provide above.

Response: Thank you for the advice. We cited the paper (43th reference paper) and added the following to the discussion section (lines 512–516):

"Second, Enformer primarily captures gene expression determinants in promoter regions while largely overlooking distal enhancers⁴³, whose activities are often cell-type-dependent and important in regulation. Because ctPred relies on Enformer predictions as input, it inherits this limitation—missing molecular information in some cell-type-specific enhancer regions—which may impact its predictive performance."

1c. It would be very informative and helpful to readers of this paper and potential users of scPredXcan to understand how the limitation of Enformer affects scPredXcan. Does this limitation explain ctPred's incorrect prediction trends (i.e., anticorrelation between observed and predicted gene expression percentiles for KCN13 and DPH1 shown in Fig. 3a)? This could occur because genomic sequence promoter signals are effectively weakening for KCN13 and DPH1 but they are cell type specific genes controlled more by their respective enhancers? One kind of analysis that would be useful would be to classify genes that the authors identify and miss (that other approaches that they compare their method to identify) according to the degree that they are cell type specific. Another interesting way to evaluate how scPred is effectively working is to perform input feature importance analysis. This can be done with DeepLift or by calculating Shapley values. Do relevant cell type specific input features (e.g., CAGE or epigenetic features from T-cells) explain model predictions (i.e., gene expression ranks for T-cells)? This analysis would only have to be performed for one of ctPred's models to be informative.

Response: Yes, the limitations of Enformer are directly inherited by ctPred, yielding incorrect prediction trends. As discussed in the paper Sasse, A. et al. "Benchmarking of deep neural networks for predicting personal gene expression from DNA sequence highlights shortcomings" Nat. Genet. 55, 2060–2064 (2023), Enformer has a large proportion (~50%) of genes with negative correlations to the ground truth. When Sasse, A. et al. dug into the details, they found that many of the single-nucleotide variant's (SNV) ISM values calculated from Enformer have signs inconsistent with measured eQTL effect sizes, which could explain why the Enformer predictions were negatively correlated with the observations at 'molecular level'. However, the underlying reasons why Enformer has this limitation remain unknown, and we and other researchers in this field are still working on addressing this issue.

The enhancer's regulation can be another influential factor limiting ctPred's prediction accuracy. Since ctPred uses the Enformer predictions for TSS regions as the input, it missed some of the distal enhancers information, which are more likely to be cell-type-specific. Performance of the model would likely improve if distal enhancers were included. As you suggested, we added the Shapley value analysis for ctPred model predicting gene expressions in T-cell (Supplementary fig. 10). We found that T-cell-related epigenomic features predicted by Enformer tend to have higher impact (absolute SHAP values) on the model prediction than other features, which indicates that ctPred uses the cell-type-specific epigenomic features for cell-type-specific gene expression prediction. Leveraging the enhancer region's molecular information can be a future direction for improvement of models like ctPred.

We a) added the Shapley value analysis to the section entitled 'ctPred accurately predicts single-cell pseudobulk gene expression across the genome in diverse cell types and datasets' (Results, lines 139–146) and b) discuss the limitations inherited from Enformer in more detail (Discussion, lines 503–507 and 512–516).

- a) *"To investigate the impact of epigenomic features on ctPred's model predictions, we conducted Shapley value analysis on ctPred trained on T-cells from the Tabula Sapiens dataset (Supplementary Fig. 10). Among the top 10 most impactful features, seven were immune related. Furthermore, across all epigenomic features, T-cell-specific features exhibited higher SHAP values (i.e., absolute impact) on model predictions compared to non-T-cell features. This finding highlights that ctPred prioritizes cell-type-specific epigenomic features to enhance the accuracy of cell-type-specific gene expression predictions."*
- b) *"First, ctPred uses Enformer as its input, and some of its predictions correlate negatively with observed expression levels due to challenges inherent to Enformer. This issue is not unique to Enformer, but represents a common limitation across current sequence-based deep learning models that predict molecular phenotypes from reference genomes^{17,18}. ... Second, Enformer primarily captures gene expression determinants in promoter regions while largely overlooking distal enhancers⁴³, whose activities are often cell-type-dependent and important in regulation. Because ctPred relies on Enformer predictions as input, it inherits this limitation—missing molecular information in some cell-type-specific enhancer regions—which may impact its predictive performance."*

Also see the figure below:

Supplementary fig. 10: Shapley value feature importance analysis of ctPred in T-cell.

Supplementary Fig. 10 a) SHAP summary plot of top10 features for ctPred in predicting gene expression percentiles in T-cell trained from Tabula Sapiens dataset. **b)** Descriptions of the top 20 impactful features for ctPred in predicting gene expression percentiles in T-cell trained from Tabula Sapiens dataset. **c)** Bar plot of mean absolute Shapley values for all the input features of ctPred in predicting gene expression percentiles in T-cell trained from Tabula Sapiens dataset. Blue: epigenomics features from T-cell. Grey: other epigenomics features.

2. The input-output relationships of ctPred and linear-ctPred should be made more clear in the main text and Figure 1. Specifically, I would make more clear in the main text that ctPred takes features derived from Enformer as input and is trained to predict cell type specific gene expression ranks/percentiles as output. Along these lines, I would remove the red arrows and "MLP model training" in the top panel (Step 1) of Fig. 1 and add an arrow to something (text or graphic) indicating gene expression ranks/percentiles as output to ctPred. If the authors want, I suggest adding the MLP model training in red under the ctPred graphic. Similarly, in the second panel (Step 2) of Fig. 1, I would remove ctPred and the violin plots and have the Enformer features going directly to linear-ctPred graphic and add an arrow to the output of linear-ctPred which points to something indicating that its output is ctPred derived gene expression ranks/percentiles. Again, if the authors wish, they can add in red below the linear-ctPred graphic the text "Elastic net training".

Response: Thanks for the suggestions. To make the training process clearer, we added the notations of 'training input' for Enformer output epigenomic features and 'training output target' for observed gene expression percentiles in grey, and added 'percentiles' after 'gene expression'. With descriptions on the left, the three steps should be clear and easy to understand.

We also changed the text in multiple places in the Overview section (lines 77–78, 82–83, and 92–95) to help clarify:

“First, we trained a model to predict gene expression percentiles from epigenomic data and observed cell-type-specific gene expression.”

“In the first step, we established a method, ctPred, to predict the cell-type-specific gene expression from individual DNA sequences”

“we used the genetically determined epigenomic features from Enformer to train ctPred—a lightweight, four-layer multi-layer perceptron (MLP) that generates single-cell expression data as output.”

See the figure below:

Fig. 1: Overview of scPrediXcan framework

Figure 1. Overview of scPrediXcan framework.

Minor comments:

1. The authors should remove text where they frame their method as a form of "transfer learning". To my understanding, transfer learning entails starting with a pretrained model whose weights have been frozen as a starting point for further model training/fine tuning. This is opposed to starting with random weights. This approach is particularly useful when there is abundant related/relevant data for a specific task for which there is limited data. The authors simply use the output of Enformer, not its model architecture which they further fine-tune with their own training data. If the authors disagree, they should cite a paper that defines the approach they take as transfer learning as opposed to what I'm detailing above. I believe keeping the framing of their method as "transfer learning" will confuse machine learning experts.

Response: Our definition of “transfer learning” follows the term “feature extraction” in “Genomic Language Models: Opportunities and Challenges”, Benegas et al., arXiv:2407.11435. However, to make our manuscript more clear to a broad audience, we have removed the reference to transfer learning and opted to explain briefly what we did.

2. In Fig. 3 legend, "GReX of in different..." should be "GReX in different...".

Response: We fixed the issue, thanks for pointing it out.

3. Should the reference in line 403 be to Fig. 6b?

Response: Yes, thanks for pointing it out.

4. In line 572, I would remove "across all individuals" as this is confusing because you already averaged them across individuals.

Response: We fixed the issue, thanks for the suggestion.

5. In the Methods section, I would replace the inaccurate reference to the output of ctPred as being "gene expression level" with "gene expression ranks/percentiles" or "gene expression percentiles". A good example of this is in line 583, but it occurs in other places. Supplementary fig. 2 should also be corrected along these lines: replacing "values" with "percentiles". I would also consider renaming the axes of plots where "Observed expressions" and "Predicted expressions" is used and add "percentiles". This is the case throughout the manuscript where these scatter plots are presented.

Response: Thanks for the suggestion, we added “percentiles” to “gene expression” in the figure legends and main texts for those ctPred scatter plots results.

6. In line 623, should you add "using a normal distribution" after "from the z-score"? How did you calculate a p-value from a z-score? Presumably via a normal distribution.

Response: That's correct. We added the content, thanks for the suggestion.

7. In Supplementary fig. 7d, the y-axis is mislabeled and should be the same as panel b.

Response: We fixed the issue, thanks for the suggestion.

Referees' report, second round of review

Reviewer 1:

The authors have addressed all my comments.

Reviewer 2:

The authors have addressed the reviewers' feedback and improved the clarity and

rigour of the study. The proposed scPrediXcan framework integrates deep learning and single-cell data to advance cell-type-specific transcriptome-wide association studies (TWAS). The methodology is innovative and the results are compelling.

Reviewer 3:

The authors have comprehensively addressed all my comments.

Authors' response to the second round of review